# Sketch-to-Skill: Bootstrapping Robot Learning with Human Drawn Trajectory Sketches

## Abstract

Training robotic manipulation policies traditionally requires numerous demonstrations and/or environmental rollouts. While recent Imitation Learning (IL) and Reinforcement Learning (RL) methods have reduced the number of required demonstrations, they still rely on expert knowledge to collect high-quality data, limiting scalability and accessibility. We propose Sketch-To-Skill, a novel framework that leverages human-drawn 2D sketch trajectories to bootstrap and guide RL for robotic manipulation. Our approach extends beyond previous sketch-based methods, which were primarily focused on imitation learning or policy conditioning, limited to specific trained tasks. Sketch-To-Skill employs a Sketch-to-3D Trajectory Generator that translates 2D sketches into 3D trajectories, which are then used to autonomously collect initial demonstrations. We utilize these sketch-generated demonstrations in two ways: to pre-train an initial policy through behavior cloning and to refine this policy through RL with guided exploration. Experimental results demonstrate that Sketch-To-Skill achieves ∼96% of the performance of the baseline model that leverages teleoperated demonstration data, while exceeding the performance of a pure reinforcement learning policy by ∼170%, only from sketch inputs. This makes robotic manipulation learning more accessible and potentially broadens its applications across various domains.

## 1 Introduction

Robots are increasingly being deployed in dynamic environments, where they must perform a wide range of tasks with precision and adaptability. One of the key challenges in enabling robots to learn new skills lies in specifying complex, task-specific behaviors. Learning from Demonstration (LfD) (Billard & Grollman (2013)) has become a widely used approach, allowing robots to acquire novel motions by imitating expert-provided trajectories. However, collecting demonstration data for LfD is challenging, particularly for high degree-of-freedom (DOF) robots performing manipulation.

Traditional methods such as kinesthetic teaching and teleoperation while useful also have challenges with safety risks, scalability, and the need for specialized expertise (Chan et al., 2014; Ferraguti et al., 2015; Bimbo et al., 2017). Recent approaches, such as using manually-operated grippers instrumented with smartphone apps (Shafiullah et al., 2023) and Virtual Reality (VR) based teleoperation systems (Kamijo et al., 2024), offer more intuitive hardware interfaces for collecting demonstrations. However, they require specialized hardware which may limit their flexibility and accessibility. Recently there has been interest in leveraging an innate human ability to communicate spatial ideas and motions through simple sketches. For example, a quick sketch of a path can easily communicate the intended movement for navigating toward a goal location.

Researchers have begun to explore this promising direction. RT-Trajectory (Gu et al., 2023) introduced the notion of sketches and showed how to use coarse trajectory sketches for policy conditioning in Imitation Learning (IL). RT-Sketch (Sundaresan et al., 2024) extended this concept to leverage hand-drawn sketches of the entire environment for goal-conditioned IL. These methods demonstrated the potential of utilizing sketches in robotics, but they were primarily focused on IL and biased towards tasks they were specifically trained on. Zhi et al. (2023) expanded this idea with *diagrammatic teaching*, where users instruct robots by sketching motion trajectories directly on 2D images of the scene. Their approach uses density estimation and ray tracing to reconstruct

Step 1: take two photos of the task scenario and collect task instructing sketches

Step 2: convert 2D sketches into 3D trajectories through pretrained generator

Step 3: follow generated trajs and perform open-loop servoing to collect experience data

Step 4: learn manipulation policy through BC warm up and demo-bootstrapped RL

Figure 1: Learning a new skill in the SKETCH-TO-SKILL framework. Step 1: Capture the task scenario from two views and collect human-drawn sketches. Step 2: Convert 2D sketches to 3D trajectories using a pretrained generator. Step 3: Execute generated trajectories to collect experience data. Step 4: Learn manipulation policy using reinforcement learning bootstrapping from behavior cloning and using guidance for experience data.

3D trajectories from the sketches, thus limiting its ability to replicate only the provided sketches and restricting its generalization to new or unseen task setups.

Unlike prior work that used sketches only as conditioning in IL, we present a more generalizable approach that learns to predict 3D trajectories from sketches in Reinforcement Learning (RL). Specifically, we propose SKETCH-TO-SKILL (Figure 1), a framework that bootstraps and guides RL using sketches. Our approach first learns to map 2D sketches to 3D trajectories, which are then used to collect demonstrations. We utilize these sketch-generated demonstrations in two ways: first, by pretraining an initial policy through Behavior Cloning (BC), and second, by refining this policy through reinforcement learning with guided exploration. Although sketch-generated demonstrations are not as precise or high-quality as teleoperated ones, they still contain enough useful information to aid RL and reduce learning time.

Unlike teleoperation which requires specialized hardware and proficiency in using the system, sketches can generally be provided by non-robotics experts. By treating these sketch-based trajectories as approximate guiding signals rather than high-fidelity demonstrations, we allow the agent to learn more effectively even with coarse sketches. We summarize our contributions as follows:

(1) We identify and address a crucial gap by integrating sketches into RL, extending their application beyond imitation learning and policy conditioning.

(2) We propose SKETCH-TO-SKILL, a framework that leverages sketches to bootstrap and guide RL, reducing reliance on high-quality, real-world demonstrations.

(3) Through extensive experiments, we demonstrate that sketches, despite their low fidelity, significantly accelerate learning by improving exploration and task comprehension in RL. SKETCH-TO-SKILL achieves $\sim$96% of the performance of the baseline model utilizing high-quality teleoperation demonstrations, while exceeding the performance of a pure reinforcement learning policy by $\sim$170% during evaluation.

## 2 RELATED WORK

**Learning from Demonstration (LfD).** LfD (Billard & Grollman, 2013) is a key method in robot learning, allowing robots to acquire skills through expert demonstrations, bypassing the complexities of action programming and cost function design (chaandar Ravichandar et al., 2020). Kinesthetic teaching, where an expert physically guides the robot while its movements are recorded, is widely used in methods like DMPs (Kober & Peters, 2009; Ijspeert et al., 2013), Probabilistic Movement Primitives (Paraschos et al., 2015), and stable dynamical systems (Khansari-Zadeh & Billard, 2011; Mohammad Khansari-Zadeh & Billard, 2014; Bevanda et al., 2022). However, it is labor-intensive and challenging to scale. Teleoperation (Si et al., 2021), where users control robots remotely, offers more flexibility but can be complex and requires expertise to operate. VR interfaces (Zhang et al., 2018; Kamijo et al., 2024) provide a more immersive alternative but depend on specialized hardware. To overcome these limitations, recent research has introduced more accessible approaches, like sketch-based demonstrations (Drolet et al., 2024).

**Sketches in Robotics.** Sketches have become a powerful tool in computer vision, aiding tasks like scene understanding (Chowdhury et al., 2023b) and object detection (Chowdhury et al., 2023a; Bhunia et al., 2023). RT-Sketch (Sundaresan et al., 2024) first explored hand-drawn sketches for goal-conditioned imitation learning (IL), using them to define tasks intuitively. RT-Trajectory (Gu et al., 2023) extended this by using trajectory sketches as IL policy conditioning, either drawn by users or generated by a Large Language Model from task descriptions. Similarly, the Diagrammatic Teaching framework (Zhi et al., 2023) uses density estimation and ray tracing to reconstruct 3D trajectories from the sketches. These methods, however, only use sketches as conditioning for task completion, and thus do not generalize beyond the tasks where the sketches are provided.

**Demonstration-Enhanced Strategies for Efficient RL.** Incorporating demonstration data in RL can improve sample efficiency, especially in environments where rewards are sparse. Methods such as Reinforcement Learning from Prior Data (RLPD) (Smith et al., 2022), Imitation Bootstrapped RL (IBRL) (Hu et al., 2023) and NAVINACT (Bhaskar et al., 2024) take advantage of prior demonstrations by embedding them into the agent's replay buffer. During training, these examples are oversampled, offering the agent more frequent exposure to expert-guided trajectories. Such approaches significantly improve learning speed and performance, particularly in continuous control tasks where learning from scratch can be prohibitively slow and inefficient (Yu et al., 2024). Our research expands upon these techniques by exploring how sketch-based trajectories can be used as an additional source of prior data in RL.

## 3 SKETCH-TO-SKILL

Our approach bootstraps robot learning from trajectory sketches, significantly lowering the barrier to entry for robotic task specification. This section details our three-stage method: (1) training a Sketch-to-3D Trajectory Generator, (2) obtaining 3D trajectories and execution experiences through the Generator and open-loop servoing, (3) pre-training an initial robotic manipulation policy through behavior cloning, and refining the policy through reinforcement learning with guided exploration. By integrating intuitive human input with powerful learning algorithms, our approach aims to create more accessible and adaptable robotic learning systems.

### 3.1 SKETCH-TO-3D TRAJECTORY GENERATOR

Our method begins with a Sketch-to-3D Trajectory Generator, $\boldsymbol{T}$, that translates a pair of 2D sketch images $(I^1, I^2)$ obtained from different viewpoints into corresponding 3D robot trajectory $\xi_g$. To train this generator, we use a dataset consisting of 3D robot end-effector trajectories along with their 2D sketches from two viewpoints. These trajectories can be obtained from various sources, such as play data where the robot executes sequences of actions. Sketches during inference can be provided by a human on RGB images of the scene, as shown in Figure 4. However, the sketches fed as input to the generator are 2D projections on blank backgrounds, with green and red dots representing the start and end points respectively, and yellow lines for the trajectory (see Figure 2 for an example). By focusing solely on the trajectory information without additional scene complexity, our model can efficiently learn to encode the dual-view sketches and decode them into the corresponding 3D trajectory.

The generator uses a neural network to map dual-view 2D sketches to 3D trajectories, where we adopt a hybrid architecture combining a Variational Autoencoder (VAE) (Kingma, 2013) and a Multilayer Perceptron (MLP), as illustrated in Figure 2. The VAE encodes sketches from two viewpoints, ideally orthogonal, to resolve depth ambiguity and capture essential trajectory features. The MLP decoder generates B-spline (Prautzsch et al., 2002) control points $C \in \mathbb{R}^{n_{cp} \times 3}$ from the latent representation, which we then use to interpolate smooth 3D trajectories. We adopted uniform knots and pre-compute the B-spline parametrization matrix $W \in \mathbb{R}^{n_{tp} \times n_{cp}}$ to reduce computational complexity and facilitate efficient backpropagation. The calculation of $W$ only depends on the uniform knot vector $\boldsymbol{u}$ and the desired number of points $n_{tp}$ in the generated trajectory, and can be pre-calculated using the Cox-de Boor recursion formula (also known as de Boor's algorithm de Boor (1977), see Appendix A for details). Then the final trajectory generation is simply a matrix multiplication: $\xi_g \in \mathbb{R}^{n_{tp} \times 3} = W \cdot C$. With the generated control point parameter $C$, we can also easily generate trajectories of varying density from the same parameters, making our method adaptable to different task requirements.

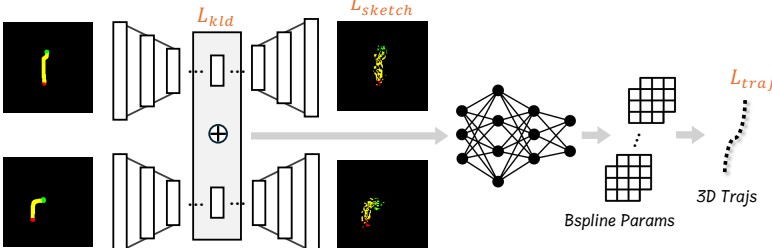

Figure 2: The Sketch-to-3D Trajectory Generator takes dual-view 2D sketches as inputs and predicts B-spline parameters to generate the final 3D trajectory output.

Our training process uses a multi-component loss function $L = L_{traj} + L_{sketch} + L_{kld}$, where $L_{traj}$ handles trajectory reconstruction, $L_{sketch}$ manages sketch reconstruction (Mean Square Error), and $L_{kld}$ is KL-divergence for latent space regularization (Figure 2). This ensures accurate trajectory generation while preserving sketch fidelity and latent space structure. We also applied data augmentation to both the sketch images and the trajectories to enhance the model's robustness and generalization (more details can be found in Appendix B).

We can use the trained Sketch-to-3D Trajectory Generator, $\boldsymbol{T}$, to generate demonstrations $\{\xi_D\}$ for learning new tasks using sketches drawn by a human. Specifically, the human draws trajectory sketches on two views of RGB images captured from the initial task state. This is similar to how human-drawn sketches are generated in prior works (Gu et al., 2023; Zhi et al., 2023). These paired sketch images, $\{(I^1, I^2)\}$, are input into our trained generator, which produces corresponding 3D trajectories, $\{\xi_g\}$, serving as the basis for guiding the robot's actions. We can also generate more than one trajectory from the same pair of sketches by adding controlled noise to the latent representation. Then we proceed to collect demonstrations for manipulation policy learning. We execute these trajectories on the robotic arm using open-loop servoing, which enables precise trajectory following based on pre-computed motor commands. During execution, we record a demonstration dataset $\{\xi_D = \{(p_t, o_t, a_t)\}_{t=1}^T\}$ at a fixed frequency, where $p_t = (x, y, z)_t$ denotes the robot's end-effector 3D position, $o_t$ represents the robot's observation, $a_t$ is the corresponding action, and $T$ is the total number of timesteps per demonstration. The collected demonstrations, which do not need to be optimal, follow the intended path while capturing the robot's actual behavior in the target environment. They serve as an effective starting point for bootstrapping the policy learning process, offering initial guidance grounded in the robot's real-world performance.

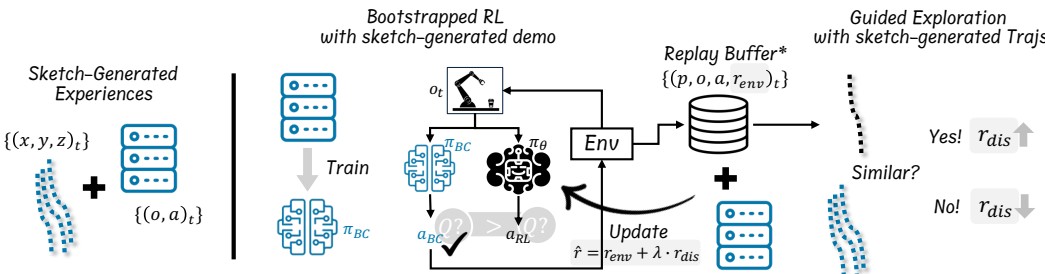

Figure 3: Overview of SKETCH-TO-SKILL integrating sketch-generated demonstrations with reinforcement learning. Sketch-generated experiences train an IL policy, which bootstraps the RL process. A discriminator guides exploration by rewarding similarity to sketch-generated trajectories. The final action, combining IL and RL policy outputs, further enhances the exploration guidance. The asterisk after "Replay Buffer" indicates that the buffer is initialized with the open-loop servoing demonstrations.

---

**Algorithm 1** SKETCH-TO-SKILL. Major modifications of IBRL highlighted in blue.

---

1: **Hyperparameters:** Number of critics $E$, number of critic updates $G$, update frequency $U$, exploration std $\sigma$, noise clip $c$, number of generated trajectories $m$ per input sketch pair, reward weighting term $\lambda$
2: **Inputs:** Pre-trained Sketch-to-3D Trajectory Generator $\boldsymbol{T}$, sketch dataset $\mathcal{S} = \{(I_i^1, I_i^2)\}_{i=1}^n$,
3: **Outputs:** Policy $\pi_\theta$, discriminator $D_\psi$

*Stage 1: Demonstration Generation*

4: $\{\xi_g\}_{1:mn} \leftarrow$ generate $m$ trajectories per sketch from $\mathcal{S}$ using $\boldsymbol{T}$
5: $\{\xi_D\}_{1:mn} \leftarrow$ generated demonstrations through open-loop servoing

*Stage 2: Policy Learning*

6: Train imitation policy $\pi_{IL}$ on demonstrations $\{\xi_D\}_{1:mn}$ using the selected IL algorithm.
7: Initialize policy $\pi_\theta$, target policy $\pi_{\theta'}$, and critics $Q_\phi$, target critics $Q_{\phi'}$, discriminator $D_\psi$ for $i = 1, \ldots, E$
8: Initialize replay buffer $B$ with demonstrations $\{\xi_D\}_{1:mn}$
9: **for** $t = 1$ **to** $N$ **do**
10:     Observe current observation $o_t$ from the environment
11:     Compute IL action $a_t^{\text{IL}} \sim \pi_{IL}(o_t)$ and RL action $a_t^{\text{RL}} = \pi_\theta(o_t) + \epsilon$, where $\epsilon \sim N(0, \sigma^2)$
12:     Sample a set $K$ of 2 indices from $\{1, 2, \ldots, E\}$
13:     Select action $a_t$ with higher Q-value from $\{a^{\text{IL}}, a^{\text{RL}}\}$
14:     Execute action $a_t$
15:     Store transition $(p_t, o_t, a_t, r_t, p_{t+1}, o_{t+1})$ in replay buffer $B$
16:     **if** $t\%U = 0$ **then**
17:         Perform discriminator $D_\psi$ update by optimizing Equation 1
18:         Perform TD3 update using minibatches from replay buffer $B$ (Fujimoto et al., 2018)
19:     **end if**
20: **end for**

---

## 3.2 POLICY LEARNING

We now describe the policy learning of the SKETCH-TO-SKILL algorithm (given in Algorithm 1). Taking as input the demonstration data $\{\xi_D\}$ collected from our Sketch-to-3D Trajectory Generator $\boldsymbol{T}$ and through open-loop serving (lines 4–5), our approach combines IL and RL to effectively bootstrap and refine the policy. Specifically, we build upon the Imitation Bootstrapped Reinforcement Learning (IBRL) framework (Hu et al., 2023), integrating our sketch-based trajectories to guide and constraint policy search space.

In IBRL we replace traditional real-world demonstrations with sketch-generated demonstrations. Initially, these sketch-based demonstrations are used to train an IL policy (line 6), which serves as a coarse approximation of the task. Although these sketches do not capture every fine detail of manipulation (e.g., gripper closing/opening actions or exact force control), our hypothesis posits that they still carry significant, actionable information that can effectively guide the learning process in reinforcement learning (RL). We leverage this information in RL in two ways (as shown in Fig. 3):

(1) Bootstrap RL with Sketch-Generated Demos: Even though sketch-generated trajectories are not as detailed as teleoperated demonstrations, they provide a foundational blueprint of the task. We leverage these initial trajectories to bootstrap our RL algorithm, giving it a preliminary direction and reducing the cold start problem common in RL scenarios. This use of imperfect demonstrations is intended to establish an initial policy that avoids random exploration at the outset, making subsequent training more focused and efficient.

(2) Guide Exploration During RL: As the agent progresses in its learning, the sketch-generated trajectories continue to serve as a guide, shaping the exploration strategy. Instead of relying on these trajectories as definitive guides, we treat them as rough outlines that suggest areas of the task space worth exploring. This guided exploration helps concentrate the agent's learning efforts on potentially fruitful regions of the action space, thus optimizing the learning speed and improving the relevance of the experiences gathered.

In both steps, the use of sketch-generated trajectories acknowledges their limitations—they are not treated as ground truth but as valuable signals to help bootstrap RL and guide exploration throughout

the learning process. For the RL algorithm, we employ TD3 (Fujimoto et al., 2018), an off-policy algorithm known for its sample efficiency. In our approach, the replay buffer is initialized with the sketch-generated demonstration trajectories (line 8), which provide an initial foundation for learning and is later updated with online experiences as the agent interacts with the environment. This combination allows the agent to refine its policy through both sketch-generated demonstration data $\xi_D$ and real-world interaction (line 13).

To further enhance the learning process and maintain consistency with the sketch-generated trajectories, we introduce a discriminator-based guided exploration mechanism (Kang et al., 2018). This discriminator, $D_\psi$, is trained to distinguish between trajectories produced by our Sketch-to-3D Trajectory Generator and those generated by the current policy:

$$\mathcal{L}_{D(\psi)} = \mathbb{E}_{p,g\sim\{\xi_D\}}[\log D_\psi(p, \Delta p, g)] + \mathbb{E}_{p,g\sim\pi_\theta}[\log(1 - D_\psi(p, \Delta p, g))], \qquad (1)$$

where $p$ represents the end-effector location, $\Delta p$ is the normalized difference between the current and next end-effector positions, capturing local trajectory characteristics, and $g$ is the task-specific information (e.g., target location). This formulation allows the discriminator to assess trajectory similarity while accounting for task variability. We then augment the TD3 reward function with an additional term based on the discriminator's output (line 18):

$$\hat{r}(o_t, a_t) = r(o_t, a_t) + \lambda \log D_\psi(p_t, \Delta p_t, g), \qquad (2)$$

where $\lambda$ is a hyperparameter controlling the influence of the discriminator. This augmented reward encourages the policy to explore state-action spaces more likely to produce trajectories similar to those generated from human sketches, potentially leading to faster learning and better performance.

Our overall learning process iterates between TD3 optimization and discriminator training. In each iteration: (1) We update the discriminator using the latest policy-generated trajectories and the original sketch-generated trajectories (line 17). (2) We then update the policy and Q-functions using TD3, with the augmented reward and guidance from the frozen IL policy (line 18). This iterative process allows the policy to refine its behavior while maintaining similarity to the initial demonstrations derived from human sketches. By combining IL, TD3, and discriminator-based guided exploration, we create a cohesive learning framework that effectively leverages sketch-based demonstrations to accelerate and improve the learning of complex manipulation tasks. Please see the Appendix for more implementation details and a complete list of hyper-parameters.

## 4 EXPERIMENTS

We report our evaluation of SKETCH-TO-SKILL, focusing on its main components: the Sketch-to-3D Trajectory Generator, the Imitation-Bootstrapped RL Policy learning, and the use of the discriminator. Our experiments address the following key questions:

Q1 How effectively does the Sketch-to-3D Trajectory Generator convert 2D sketches into usable 3D robot trajectories?

Q2 Can SKETCH-TO-SKILL utilize sketch-generated demonstrations to achieve comparable performance to traditional methods using high-quality demonstration data?

Q3 How do various design choices in SKETCH-TO-SKILL, such as the number of generated demonstrations per sketch and the discriminator reward weighting, affect the learning and refinement of robotic policy?

Q4 How well does our method translate to the real world?

### 4.1 EVALUATION OF THE SKETCH-TO-3D TRAJECTORY GENERATOR

The Sketch-to-3D Trajectory Generator is a key component of SKETCH-TO-SKILL, translating 2D sketch inputs into 3D robot trajectories. To train this generator, we collect data of the robot arm executing *play* trajectories. We record the 3D trajectories as well as their 2D projections from two viewpoints. We create such a dataset in the Metaworld (Yu et al., 2019) simulation environment as well as a separate one using actual hardware (Figure 13). Once the Sketch-to-3D Trajectory Generator is trained, we can use hand-drawn sketches as input to predict 3D trajectories.

**Performance on Hand-drawn Sketches.** We provide an example using the `ButtonPress` task to qualitatively assess the generator's effectiveness with hand-drawn inputs (Figure 4). We asked users

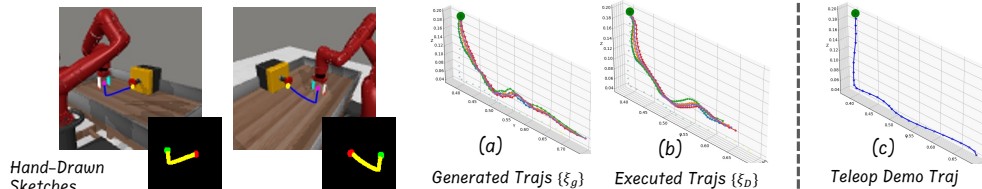

Figure 4: Multi-stage trajectory generation and execution. On the left, we show hand-drawn sketches on scenario RGB images and the extracted sketches on a blank background, (a) generated trajectory from the Sketch-to-3D Trajectory Generator, and (b) executed trajectory via open-loop serving. In (c), we visualize a teleoperated demo for the same task for reference.

to provide sketches for the task and also separately collected actual demonstrations as a reference. We see that the Sketch-to-3D Trajectory Generator was able to predict trajectories (Figure 4a) similar to the actual demonstrations (Figure 4c). We also generate more than one trajectory from the same pair of sketches by adding controlled noise to the latent representation. This approach allows us to produce a range of plausible trajectories for a given sketch input, enriching the demonstration set and potentially leading to more robust and adaptable robot policies. We then execute the generated trajectories to produce demonstrations for training the policy (Figure 4b). Despite the inherent variability in sketch inputs, the executed trajectory further validates the practical applicability of our approach. This demonstrates our model's robustness to sketch imperfections and its ability to reliably interpret user intent, bridging the gap between simple 2D sketches and actionable 3D robot trajectories.

**Latent Space Representation and Interpolation.** To further understand the generator's latent space, we performed linear interpolation in the latent space between different input samples. Specifically, we selected two distinct sketch pairs with different trajectories, extracted their feature vectors, linearly interpolated between them, reconstructed the sketches, and generated new trajectories. Figure 12 shows smooth transitions in both 2D sketches and 3D trajectories across the interpolated latent space. This smoothness demonstrates that our model has learned a continuous and semantically meaningful representation, suggesting good generalization capability to unseen inputs that lie between known examples (Kingma, 2013). The coherence between interpolated sketches and their corresponding 3D trajectories further validates the model's robust sketch-to-trajectory mapping.

## 4.2 COMPARISONS WITH BASELINES

In this section, we conduct extensive experiments in MetaWorld (Yu et al., 2019) to answer Q2: *can* SKETCH-TO-SKILL *utilize sketch-generated demonstrations to achieve comparable performance to traditional methods using high-quality demonstration data?* Specifically, we compare SKETCH-TO-SKILL with: (1) IBRL (Hu et al., 2023), a strong baseline that utilizes traditional high-quality demonstration data (rather than sketches as what our method uses), and (2) TD3 (Fujimoto et al., 2018), a state-of-the-art pure RL approach without using any demonstrations. We hypothesize that although the sketches have only partial information (namely, 2D projections of 3D trajectories and no gripper information), we can still generate good enough demonstration data to perform comparably with the baseline that uses full demonstrations. We show that to be the case in these experiments.

We perform evaluations on six tasks from the MetaWorld benchmark, namely `Coffeepush`, `Boxclose`, `Buttonpress`, `Reach`, `Reachwall`, and `ButtonpressTopdownwall`, each using sparse 0/1 task completion rewards at the end of each episode. For each task, we collected 3 high-quality demonstrations using an expert policy. These demonstrations served as our baseline for traditional demonstration-based methods. For our approach, we collected a total of three hand-drawn sketches, one on each demonstration's initial frames (Figure 4). These sketches were used to generate and execute trajectories, creating a parallel set of sketch-based demonstrations for comparison.

Figures 5 and 6 show the training and evaluation performance across all the tasks. We present SKETCH-TO-SKILL's results with and without the discriminator reward. We observe that in all cases SKETCH-TO-SKILL performs better, often significantly better than pure RL. In most tasks,

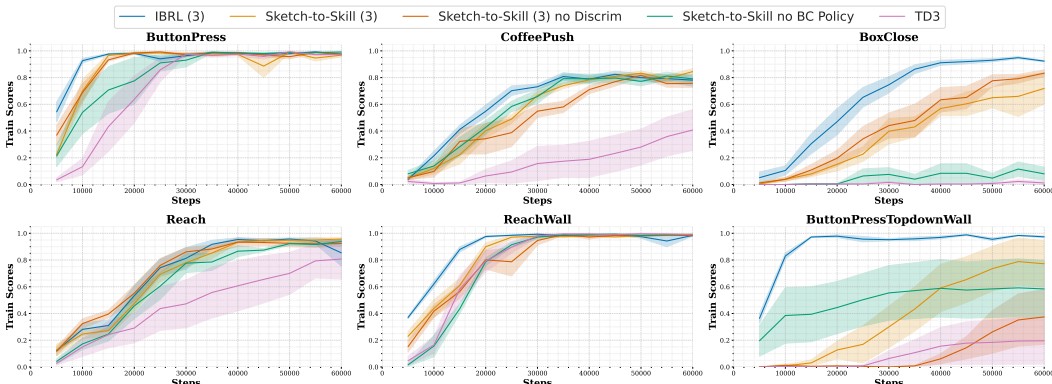

Figure 5: **Performance Comparison of SKETCH-TO-SKILL Across MetaWorld Tasks with Baselines.** This figure shows the training score (success rate) for six MetaWorld environments. SKETCH-TO-SKILL, with hand-drawn sketches, achieves comparable performance to IBRL which uses actual teleoperated demonstrations while being much better than pure RL.

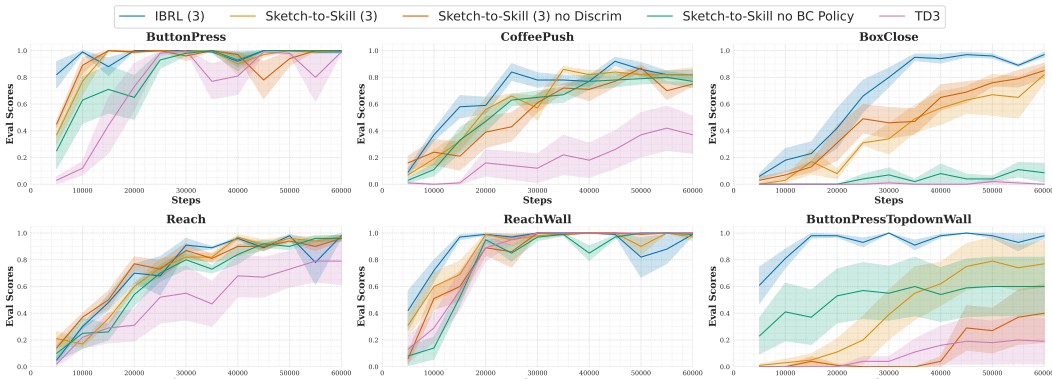

Figure 6: **Evaluation Scores.** This figure shows the evaluation score (success rate) for six Meta-World environments during evaluation.

SKETCH-TO-SKILL's performance using only sketches as input is comparable to IBRL which uses high-quality demonstrations. This is particularly notable in the `CoffeePush` and `Boxclose` tasks. These tasks require actuating the gripper — information that is not provided in the sketches. Nevertheless, SKETCH-TO-SKILL is able to bootstrap and use guidance from the sketch generated sub-optimal demonstrations to learn a policy efficiently. This provides evidence to the claim that the sketch-generated demonstrations do not lead to much degradation in performance while being much easier to obtain.

**Behavioral Cloning Performance.** SKETCH-TO-SKILL employs behavior cloning (BC) to bootstrap policy learning, similar to IBRL. However, the key difference is that IBRL relies on high-quality teleoperated demonstrations, whereas SKETCH-TO-SKILL uses sketch-generated demonstrations. We compare the performance between them (Figure 7) and ablate the number of generated demonstrations $m$ per input sketch pair. Not surprisingly the BC policy with teleoperated data performs better than the sketch generated ones. However, despite the lower performance of the BC policy, SKETCH-TO-SKILL is still able to achieve comparable performance in RL training (as seen in Figures 5 and 6), showing that it is not as sensitive to the quality of the bootstrapping policy. Increasing the number of generated demonstrations $m$ per input sketch pair (from 1 to 10) does not significantly improve the BC performance.

In `ButtonpressTopdownWall`, the sketch-generated dataset resulted in a BC policy that fails in all cases. However, as we observe in Figures 5 and 6, SKETCH-TO-SKILL that uses this policy is still able to perform better than pure RL. This can be attributed to the fact that the sketch-generated

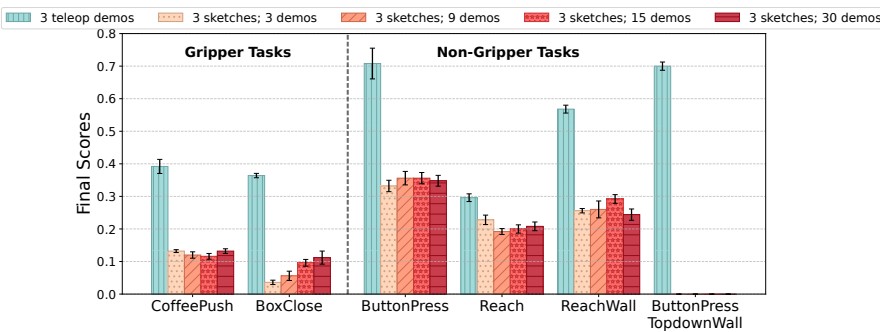

Figure 7: **Behavioral Cloning (BC) scores using actual teleoperated data and sketch generated demonstrations**. The blue bars represent the baseline BC policy trained with 3 high-quality demonstrations, while the red bars show BC policies trained with sketch-generated demonstrations, varying in the number of demonstrations $m$ per input sketch pair (1, 3, 5, and 10). Darker shades of red indicate an increase in the number of sketch-based demonstrations used for training. Despite poor success rate, the actual trajectories and policy learned with sketches are useful for bootstrapping as evidenced by the training performance (Figures 5 and 6).

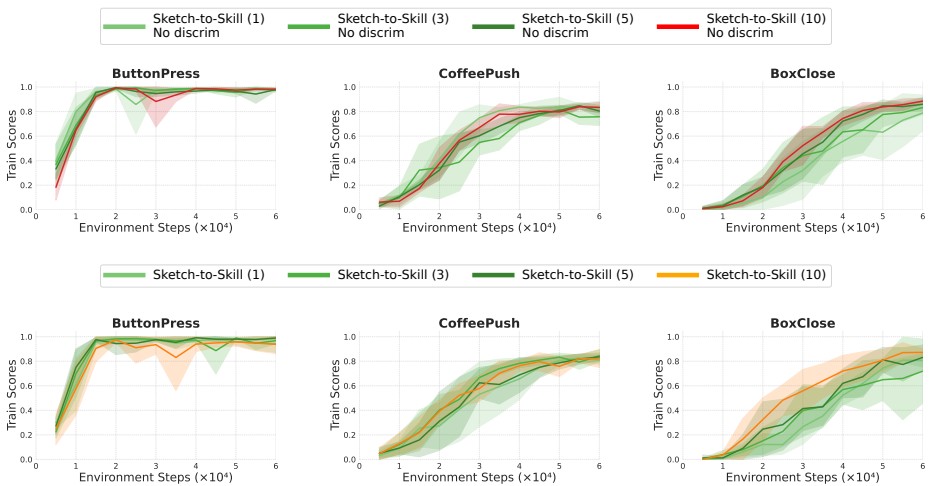

Figure 8: Top row illustrates ablation training scores for SKETCH-TO-SKILL without discriminator and the bottom row shows with discriminator. We vary $m$, the number of demonstrations generated per sketch pair (1, 3, 5, and 10).

demonstrations, due to the inherent noise, are not able to actually succeed in pushing the button. However, the trajectories themselves reach very close to the button. To verify this, we computed the final position difference between the teleoperated demonstrations and the sketch-generated ones which was 0.001. Therefore, even though the success rate of the BC policy is close to 0, the discriminator guidance and the bootstrapping allow SKETCH-TO-SKILL to learn effectively.

## 4.3 ABLATION STUDY

To understand the impact of the key components in SKETCH-TO-SKILL and answer Q3, we conducted ablation studies focusing on two critical aspects: the number of generated trajectories $m$ per input sketch pair and the reward weighting scheme $\lambda$.

**Impact of Generated Trajectories per Sketch.** We investigated how the number of trajectories generated from each input sketch pair affects the learning performance. Figure 14, shows the learning curves for policies trained with varying numbers of generated trajectories per sketch. We see that the performance is improved when we generate $m = 3$ trajectories per sketch, instead of just one

trajectory per sketch. Here, the additional demonstrations can make up for the deficiency of not having actual teleoperated demonstrations. However, increasing the number of trajectories per sketch has diminishing value. It is useful when the tasks are difficult, such as `BoxClose` and `CoffeePush` which involve gripper actions, but does not affect much for easier tasks.

**Effect of Reward Weighting**: We examined the impact of different reward weighting schemes on policy learning. Our reward function combines the environmental reward with a discriminator-based reward by Equation 2, where $\lambda$ is the weighting parameter. Figure 9 illustrates the learning performance across different values of $\lambda$. The model demonstrates comparable performance with reward weights of 0.1 and 0.005, but significantly underperforms with a reward weight of 0.5.

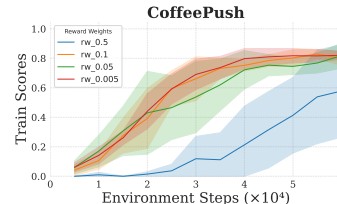

Figure 9: Reward weighting term ablation

### 4.4 HARDWARE EXPERIMENTS

We validate SKETCH-TO-SKILL on physical robot hardware to demonstrate its effective transfer from simulation to real-world applications.

**Experimental Setup.** We set up the ButtonPress task as shown in Figure 10. We use a UR3e robot equipped with a Robotiq hand-e gripper and a realsense camera mounted on the wrist. We also use two additional environmental cameras to capture frames for humans to draw sketches on (Figure 13). The details of the task, success detection, and reset mechanism are in the Appendix.

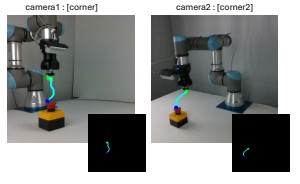 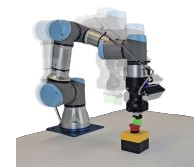 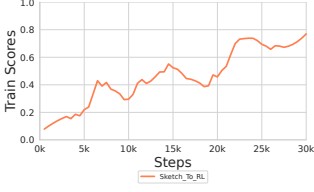

Figure 10: Real experiment training scores for SKETCH-TO-SKILL with BC success rate of 0.8.

**Performance.** The evaluation success rate of the BC policy of ButtonPress task trained on sketch-generated demonstrations is notably high at 0.8 within the randomized environment where we executed the policy. Consequently, our sketch-to-skill policy without a discriminator, quickly demonstrated strong performance, achieving a training success rate of 0.8 within just 30K samples.

## 5 CONCLUSIONS AND FUTURE WORK

We present SKETCH-TO-SKILL that uses 2D sketches to improve the efficiency of learning a manipulation skill. While prior work has shown the power of sketches in IL, we are the first to show how to do so using RL. The key ideas are to train a 2D sketch to 3D trajectory generator whose output is used to bootstrap learning of the RL policy and used as an extra exploration guidance signal, all of which contributes to improved efficiency. There are several avenues for future work. SKETCH-TO-SKILL currently does not include any gripper information or timing information in the sketches. While we have shown that even without this information, we can learn effectively, an immediate line of work would be to include this in the sketches. Fundamentally, this does not change the Sketch-To-3D trajectory generator model which would now also have to predict the gripper state and time parameterization of the trajectory. The second avenue for future work is to generate the sketches from sources other than humans. For example, Gu et al. (2023) showed how to generate sketches using a Vision Language Model given a natural language description of the task. We can directly incorporate such sketches into our framework.

## 6 REPRODUCIBILITY

Anonymized code and demo datasets will be available on our webpage. We use the standard Meta-World benchmark to allow for easy comparison with other algorithms and to facilitate the reproducing of our results. All details about the hyperparameters, environment specifications, and real-world experiment setup are provided in the appendix.

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

## A DE BOOR'S ALGORITHM DETAILS

The uniform knot vector for a B-spline of degree $p$ with $n + 1$ control points is defined as:

$$\boldsymbol{u} = [\underbrace{0, ..., 0}_{p+1}, \frac{1}{n-p+1}, \frac{2}{n-p+1}, ..., \frac{n-p}{n-p+1}, \underbrace{1, ..., 1}_{p+1}] \tag{3}$$

The B-spline basis functions $\boldsymbol{W}_{i,p}(t)$ are defined recursively using the Cox-de Boor recursion formula:

$$\boldsymbol{W}_{i,0}(t) = \begin{cases} 1 & \text{if } u_i \leq t < u_{i+1} \\ 0 & \text{otherwise} \end{cases} \tag{4}$$

$$\boldsymbol{W}_{i,p}(t) = \frac{t - u_i}{u_{i+p} - u_i} \boldsymbol{W}_{i,p-1}(t) + \frac{u_{i+p+1} - t}{u_{i+p+1} - u_{i+1}} \boldsymbol{W}_{i+1,p-1}(t) \tag{5}$$

where $u_i$ are the knot values from the knot vector $\boldsymbol{u}$.

The B-spline parametrization matrix $N$ for $m$ evaluation points is an $m \times (n + 1)$ matrix:

$$\boldsymbol{W} = \begin{bmatrix} \boldsymbol{W}_{0,p}(t_1) & \boldsymbol{W}_{1,p}(t_1) & \cdots & \boldsymbol{W}_{n,p}(t_1) \\ \boldsymbol{W}_{0,p}(t_2) & \boldsymbol{W}_{1,p}(t_2) & \cdots & \boldsymbol{W}_{n,p}(t_2) \\ \vdots & \vdots & \ddots & \vdots \\ \boldsymbol{W}_{0,p}(t_m) & \boldsymbol{W}_{1,p}(t_m) & \cdots & \boldsymbol{W}_{n,p}(t_m) \end{bmatrix} \tag{6}$$

where $t_j$ $(j = 1, ..., m)$ are evenly spaced parameters in the interval $[0, 1]$.

## B SKETCH-TO-3D TRAJECTORY GENERATOR ARCHITECTURE

### OVERVIEW OF THE MODEL

The proposed model converts 2D image sketches into 3D motion trajectories using a *Variational Autoencoder (VAE)* combined with a *Multi-Layer Perceptron (MLP)*. The VAE encoder processes $64 \times 64$ pixel 2D sketches (3 channels) into a latent vector ($d_{\boldsymbol{v}} = 32$), while the decoder reconstructs the sketches to retain essential features for trajectory generation. The latent space outputs the mean ($\mu$) and variance ($\sigma^2$), sampled using the reparameterization trick.

The MLP takes the latent vectors from two sketches, concatenates them, and generates 3D control points for B-spline trajectory interpolation. The MLP takes an input of size ($d_{\boldsymbol{v}} \times 2$), processes it through hidden layers [1024, 512, 256], and outputs $n_{cp} \times 3$. The generated 3D control points are then used for B-spline interpolation to produce smooth trajectories.

### INITIALIZATION, REGULARIZATION, AND HYPERPARAMETERS

We initialize all network parameters using *Xavier initialization*. Regularization is done with *Kullback-Leibler Divergence (KLD)*, using a loss function that combines *Sketch Reconstruction Loss*, *KLD Loss*, and *Trajectory Loss*. The *Sketch Reconstruction Loss* is the MSE loss on sketch images, the *Trajectory Loss* computes the MSE between predicted and ground truth trajectories, where the ground truth trajectory is first densely fitted and resampled to ensure uniform point spacing, and the *KLD Loss* applies Kullback-Leibler Divergence for regularization. Key hyperparameters include:

- **Image size:** $64 \times 64$ pixels
- **Latent dimension:** 32
- **Number of control points:** 20
- **B-spline degree:** 3

- **MLP hidden layers:** [1024, 512, 256]

- **Learning rate:** $1 \times 10^{-3}$ (Adam optimizer)

- **Batch size:** 128

- **KLD weight:** 0.0001 (with optional annealing)

- **Training epochs:** 200

To enhance robustness and generalization, our training process employs two concurrent data augmentation strategies. The first applies diverse image augmentations (rotations, scaling, affine transformations, noise) to input sketches, used exclusively for updating the VAE to learn robust sketch representations. The second strategy targets potential mismatches in hand-drawn sketches by subtly modifying both original sketches and their 3D trajectories. This involves adding noise and minor elastic deformations to sketches, and noise with refitting to trajectories. These augmented pairs update the entire model, preparing it for hand-drawn input variability while maintaining sketch-trajectory consistency. This augmentation approach enhances the model's ability to handle diverse, imperfect sketches while ensuring accurate 3D trajectory generation in real-world scenarios. To train the sketch generation network we collect 22000 samples for simulation tasks and 85 for the real world. During the training for the real-world task, we use the data collected from

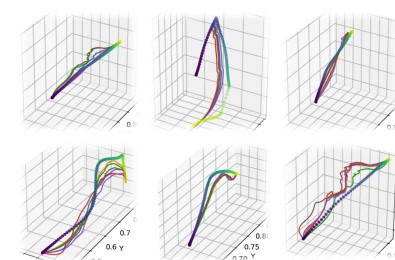

Figure 11: Diversity in generated 3D trajectories. Each subplot shows multiple generated trajectories (colored lines) for a single input, demonstrating variability. Scattered points represent the ground truth trajectory.

simulation to train the VAE part as well. We provide additional visualizations to demonstrate the learned model's generalizability in Figure 11 and 12.

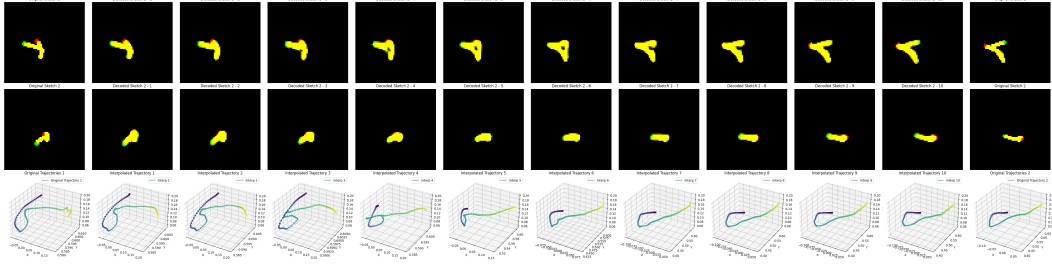

Figure 12: Latent space interpolation results showing smooth transitions between original samples (leftmost and rightmost) and reconstructed samples (middle) from two viewpoints, demonstrating the model's ability to generate coherent 3D trajectories from 2D sketch pairs.

## C  IMPLEMENTATION DETAILS AND HYPERPARAMETERS OF SKETCH-TO-SKILL POLICY AND BASELINES

This section outlines the implementation details of Sketch-To-Skill and the baselines. The behavior cloning (BC) policies utilize a ResNet-18 encoder, where the output is flattened and processed by MLPs to produce final 4D actions. We replace the BatchNorm layers in ResNet with GroupNorm, matching the number of groups to the input channels. To prevent overfitting, we employ random-shift data augmentation.

In the Meta-World environment, we utilize a corner2-image camera setup. We use wrist cameras to enhance generalization and sample efficiency in real-world experiments, specifically using them for the ButtonPress task.

Table 1: Hyperparameters for RL in SKETCH-TO-SKILL.

| Parameter | Meta-World | Real-World |
|---|---|---|
| Optimizer | Adam | Adam |
| Learning Rate | 1e-4 | 1e-4 |
| Batch Size | 256 | 256 |
| Discount ($\gamma$) | 0.99 | 0.99 |
| Exploration Std. ($\sigma$) | 0.1 | 0.1 |
| Noise Clip ($c$) | 0.3 | 0.3 |
| EMA Update Factor ($\rho$) | 0.99 | 0.99 |
| Update Frequency ($U$) | 2 | 2 |
| Actor Dropout | 0.5 | 0.5 |
| Q-Ensemble Size ($E$) | 2 | N/A |
| Num Critic Update ($G$) | 1 | N/A |
| Image Size | 96×96 | 96×96 |
| Use Proprio | No | N/A |
| Proprio Stack | N/A | N/A |
| State Stack | N/A | N/A |
| Action Repeat | N/A | N/A |

## D  ADDITIONAL DETAILS OF REAL-WORLD EXPERIMENTS

In this section, we present insights into the real-world experiments conducted using Sketch-To-RL, specifically for the ButtonPress manipulation task. We utilized a UR3e robot equipped with a Robot Hand gripper, operating in an action space with 4 dimensions: 3 for end-effector position deltas under a Cartesian impedance controller and 1 for the absolute gripper position, with policies functioning at 7.5 Hz.

To train the Sketch-To-3D Trajectory generator, we collected approximately 85 teleoperated trajectories. RGB images were captured using two orthogonally positioned RealSense cameras to enhance trajectory insight. A green marker was placed on the gripper tip to facilitate sketch generation on the frames, enabling the model to learn 2D sketch projections onto 3D trajectories.

After training the Sketch-To-3D Trajectory generator, we created 30 sketches based on RGB frames from the two cameras. We then collected 30 sketch-generated demonstrations, $\xi_D$ using openloop servoing on 3D trajectory $\xi_g$ produced by the generator $\boldsymbol{T}$. We then train Behavior Cloning (BC) policy using the sketch-generated demonstrations, $\xi_D$, achieving a score of 0.8 and thus leading to good performance in the sketch-to-skill policy.

All methods maintained the same hyperparameters and network architectures as those used in the Meta-World tasks. The task is illustrated in Figure 10 and briefly described below:

**ButtonPress:** The objective is to press the Button, with its initial position randomized within a 20cm by 20cm to 25cm trapezoidal area visible from the wrist camera. We collected 30 demonstrations for this task.

**Reward Detection :** We used a manual method to reward the agent if it successfully pushed the button. The agent receives a reward of 1 for successfully completing the task and a reward of 0 in all other cases. Each episode has a set number of timesteps, and if agent doesn't succeed within that limit, it resets and starts over. The length of each episode may be shorter than the given limit depending on how quickly the agent completes the task.

**Reset :** For the object, we manually reset the environment by pulling back the button (if pressed) and uniformly randomizing the position of the button in each episode. The robot is initially set to a specific joint configuration known as the home position. Whenever the agent receives a reward or an episode ends, it resets back to the home configuration and the training continues. The object randomization is done by placing the button at the center, and the position remains unchanged until the agent receives its first reward. After that, the object is gradually moved towards the boundary, circled around the workspace, returned to the center, and the process is repeated until the training ends.

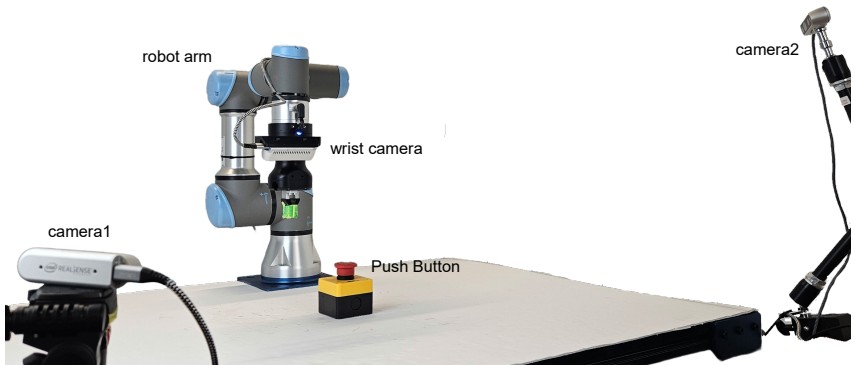

Figure 13: Complete setup for the ButtonPress task in a real-world experiment. The configuration includes a UR3e robot arm equipped with a Robot Hand gripper, and a RealSense D435i camera mounted on the wrist. Two additional RealSense cameras are positioned orthogonally to capture the trajectory from two different viewpoints.

**Safety Boundaries:** We have restricted the movement of the robot to the x, y, and z directions of the end-effector. Each step that the robot takes is limited to a specific value. If the action taken by the agent exceeds this limit, the robot will not move and will remain in its current position. To avoid collision after pressing the button and pushing it into the table, we have set a minimum limit for the z direction of the robot, ensuring it can still press the button. Even if the agent attempts to move downward into the table, the robot will remain at the specified z position threshold.

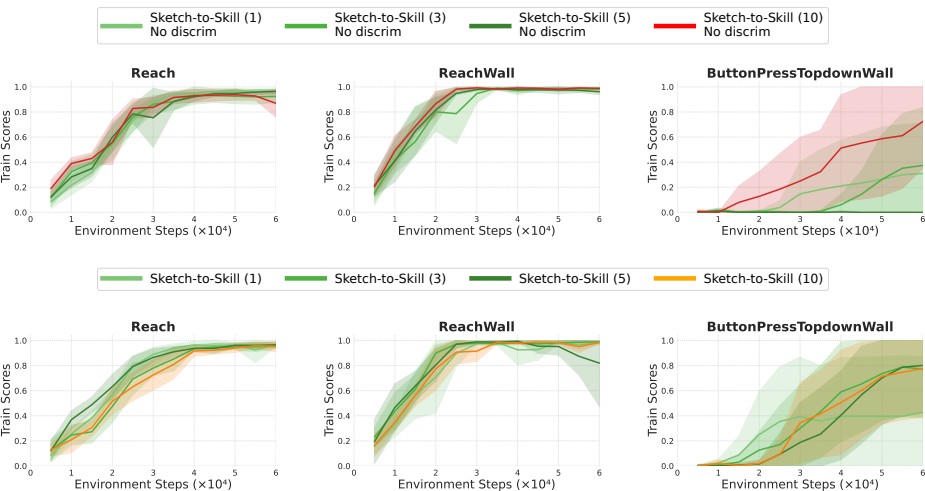

Figure 14: Top row illustrates ablation training scores for SKETCH-TO-SKILL without discriminator and the bottom row shows the ablation training scores for SKETCH-TO-SKILL, trained with demonstrations generated from sketches for bootstrapping, varying in the number of generated demonstrations $m$ per input sketch pair (1, 3, 5, and 10).

# E    ADDITIONAL EXPERIMENTS FOR REBUTTAL

## E.1    AE1: ROBOMIMIC EXPERIMENTS

### E.1.1    TASK COMPLEXITY:

The PickPlaceCan task in RoboMimic is another demanding two-stage challenge. It requires the robot to accurately locate and reach a can, then pick it up and place it into a designated bin. This

task not only tests the robot's ability to handle objects with precision but also demands correct orientation of the gripper throughout the process. RoboMimic, a well-established benchmark, provides high-quality demonstrations collected via human teleoperation, which are instrumental for training successful policies.

### E.1.2 IMPLEMENTATION AND STRATEGY:

In this setup, the sketch-based demonstrations provide initial positional cues for the robot. Our framework utilizes reinforcement learning (RL) to refine these initial cues, dynamically adjusting the robot's approach to manage both the orientation and timing necessary for successful task execution. This method proves particularly effective as it does not rely on fully detailed trajectory information from the start; instead, it uses the sketches to guide the initial exploration phase of RL, significantly simplifying the data collection process.

**Results and Performance Metrics:**

- **Performance:** Our sketch-to-skill framework, even with limited initial data, performs admirably in this demanding scenario. The results are on par with those from IBRL methods 98%, which benefit from complete and detailed human demonstrations, including explicit orientation details (as shown in Figure 15).

- **Comparison to IBRL:** The comparative success illustrates the robustness and effectiveness of our method in managing the task's orientation and other complexities without fully specified trajectory inputs. This is particularly notable given that RoboMimic's Pick and Place Can task offers a significantly higher level of difficulty than similar tasks in Meta-World.

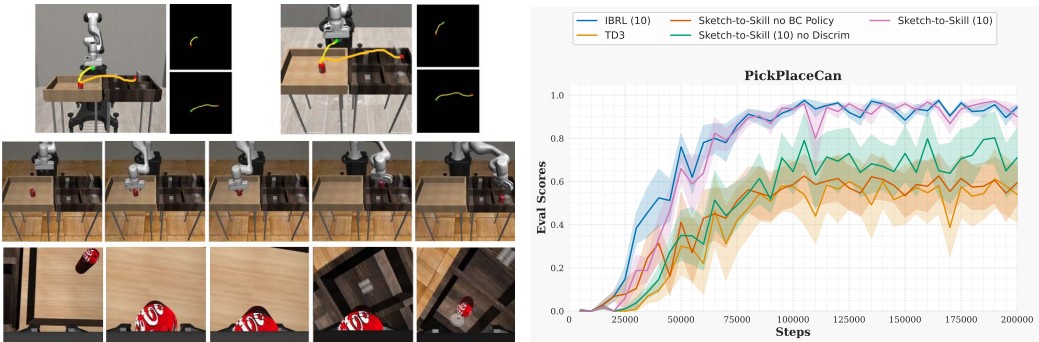

Figure 15: Evaluation Scores (success rate) for the robomimic PickPlaceCan environment during evaluation.

### E.2 AE2: HARDWARE EXPERIMENTS

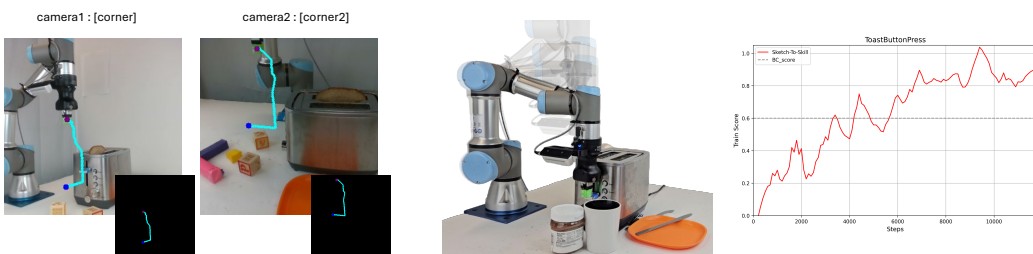

Figure 16: Training curves of the real-world ToastButtonPress task, where the environment is cluttered

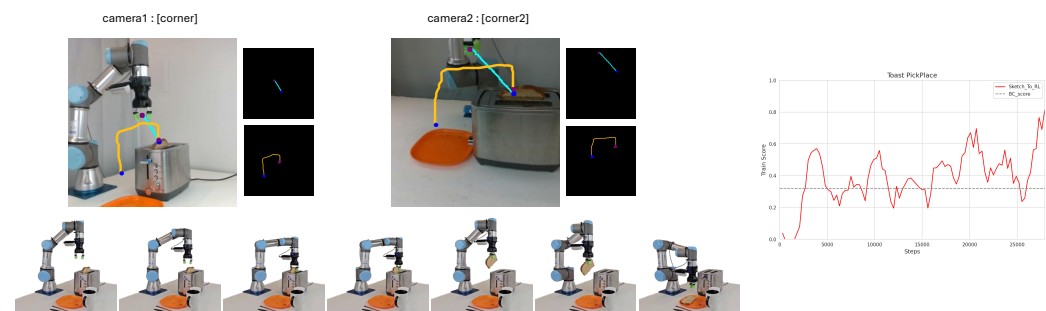

Figure 17: Training curves of the real-world ToastPickPlace task, where the environment is cluttered

This appendix provides detailed insights into additional real-world experiments conducted using the Sketch-To-RL framework, focusing on the Toaster Button Press and Bread Pick Place tasks. Both experiments were carried out using a UR3e robot equipped with a Robotiq Hand gripper, navigating an action space that included three dimensions for end-effector position deltas and one for absolute gripper position. The policies operated at 7.5 Hz. These experiments were designed to test the robustness and adaptability of our sketch-based approach under dynamic, cluttered environments with randomized initial conditions.

### E.2.1 TOASTER BUTTON PRESS EXPERIMENT

- **Sketch and Demonstration Generation:** Created 10 sketches based on the camera feeds. From these, 10 sketch-generated demonstrations were collected using open-loop servoing based on 3D trajectories produced by the generator which was previously trained on 85 trajectories collected earlier.

- **Experimental Design:** Each episode featured a cluttered environment around the toaster, with objects commonly found in household settings to simulate realistic conditions. The initial position of the gripper was randomized in every episode to test the adaptability of the learned policy.

- **Training and Performance** Trained a Behavior Cloning (BC) policy using the sketch-generated demonstrations, achieving a preliminary success rate of 60%, and trained Sketch-To-Skill over 12k interaction achieving 90% within 10K interactions.

### E.2.2 BREAD PICK PLACE EXPERIMENT

- **Setup and Data Collection:** For this task, sketches were specifically collected in randomly cluttered environments to reflect typical variability in real-world scenarios.

- **Experimental Design:** The task involves picking a piece of bread from a toaster and placing it on a nearby plate, requiring precise manipulation and handling. Both the environment clutter and the initial gripper positions were randomized in each episode, presenting a different challenge each time to test the robustness of the policy.

- **Training and Performance:** The BC policy was similarly trained using sketch-generated demonstrations, with performance metrics collected to assess the effectiveness of the approach under varied and dynamic conditions achieving success of 36% and trained Sketch-To-Skill over 30K interaction achieving 80% within 30 interactions.

### E.2.3 COMMON ELEMENTS ACROSS EXPERIMENTS:

**Network Architecture and Hyperparameters:** All methods maintained consistent network architectures and hyperparameters as used in the Meta-World tasks.
**Reward Detection and Reset Protocol:** A manual reward detection method was employed, where the robot received a reward for successfully completing the designated task. The environment and robot position were reset at the end of each episode to ensure consistent training conditions.

These experiments underscore our method's capability to handle real-world variability and complex task execution, supporting its potential utility beyond controlled experimental setups. The detailed results from these tasks, illustrated in Figures 16 and 17, highlight the practical applications and adaptability of our sketch-to-skill framework in dynamic, cluttered settings.

### E.3 AE3: Hard Metaworld Experiments

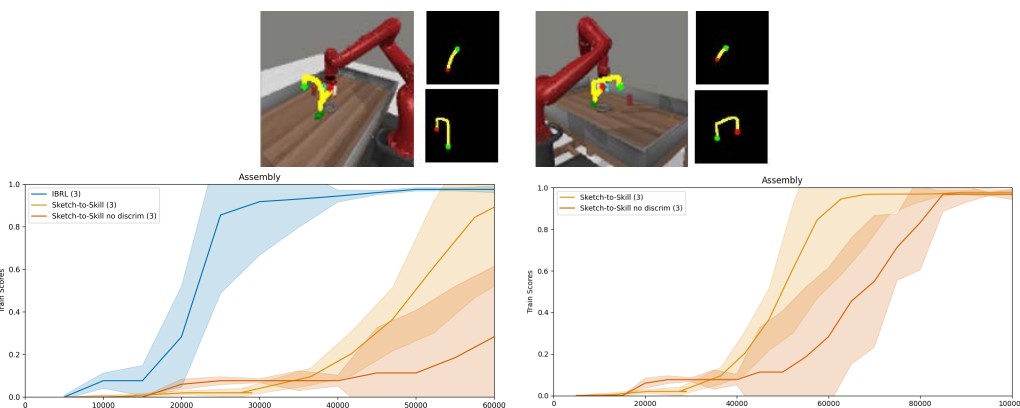

Figure 18: Figure on top shows sketches for the MetaWorld Assembly task. Figures on the bottom show training scores (success rate).

MetaWorld Assembly Task Description

**Task Complexity:** The Assembly task in MetaWorld is a "hard" task Seo et al. (2023) that requires the robot to execute two-stage manipulations. The task involves precise movements to pick up a peg, navigate it to a specific location, and insert it into a hole within a larger assembly fixture. This task tests the robot's precision, spatial awareness, and ability to handle complex sequences.

**Results and Performance Metrics:**

- Sketch-To-Skill: Demonstrated a high success rate of 93%, effectively using sketches for coarse guidance to navigate and complete complex task sequences, even without actual teleoperated demonstrations.

- Sketch-to-Skill without Discriminator: Initially struggled with a success rate of 20%, but extended interaction up to 100K steps improved performance dramatically, reaching a near-perfect success rate of 98%. This underscores the potential for learning even without discriminator guidance, given sufficient training time.

- IBRL: Achieved the highest success rates of approximately 100%, benefiting from high-quality teleoperated demonstrations that include precise details on orientation and positioning.

- Standard BC: Achieves a success rate of around 60%, showing limitations in environments where adaptive behaviors and fine-tuning through reinforcement learning are necessary.

These results highlight the effectiveness of the Sketch-to-Skill approach in handling complex, multi-stage tasks through initial coarse guidance, with the potential for significant improvement over time. They also illustrate the advantage of incorporating discriminator feedback to accelerate learning and enhance performance.

### E.4 AE4: Use of VAE in Sketch-to-3D Trajectory Generator

We conducted an ablation study of the Sketch-to-3D Trajectory Generator to evaluate the effect of the VAE in the architecture. We used 1000 trajectories, split 80:20 for validation and report the training and validation loss in Table 2. We compare the performance of using the VAE in Figure 2

versus without the VAE and using an CNN (with the same architecture as the VAE's encoder, minus the decoder and loss components) instead. We observe that the VAE improves the performance across all the losses including the reconstruction loss and trajectory loss.

Table 2: Performance Metrics for generator model

| Metric | with VAE | without VAE |
|---|---|---|
| Training Loss | **0.6019** | 0.6286 |
| Validation Loss | **0.9128** | 1.2804 |
| Reconstruction Loss | **0.0004** | 0.3258 |
| KLD Loss | **69.8592** | 107.167 |
| Parameter MSE Loss | **0.0820** | 0.0928 |
| Trajectory Loss | **0.770** | 0.1180 |

### E.5 AE5: Using Color Gradients for Overlapping Trajectories

We can also incorporate time-parameterization of the trajectory in the sketches using a color gradient, instead of a binary sketch image. This notion of color gradient in sketches was introduced by RT-Trajectory Gu et al. (2023). An example is shown in Figure 19. Here, the color of the sketch changes from green (start) to red (end). This is particularly helpful when the sketch crosses each other or overlaps. We conduct additional experiments with the generator to evaluate the effect of incorporating gradients in the Sketch-To-3D trajectory generator.

Specifically, we use the MetaWorld Assembly task (Figure 18) where the sketch overlaps in the middle as the end-effector picks up the tool and carries it to the goal position. We trained two generators without and with color gradients. The performance of these two generators are reported in Table 3. We observe that incorporating the gradient in such a case results in lower reconstruction and trajectory losses. With lower trajectory losses we can handle more complex trajectories that overlap with color gradients. Note that incorporating gradients also does not require any change to the downstream architecture, and only requires minimal changes to the generator architecture.

Table 3: Performance Metrics for Overlapping and Non-Overlapping Trajectories

| Metric | with color gradient | without color gradient |
|---|---|---|
| Training Loss | **0.2438** | 0.3107 |
| Validation Loss | **0.3120** | 0.3585 |
| Reconstruction Loss | **0.0001** | 0.0003 |
| KLD Loss | **77.167** | 76.4683 |
| Parameter MSE Loss | **0.1028** | 0.1135 |
| Trajectory loss | **0.1530** | 0.2379 |

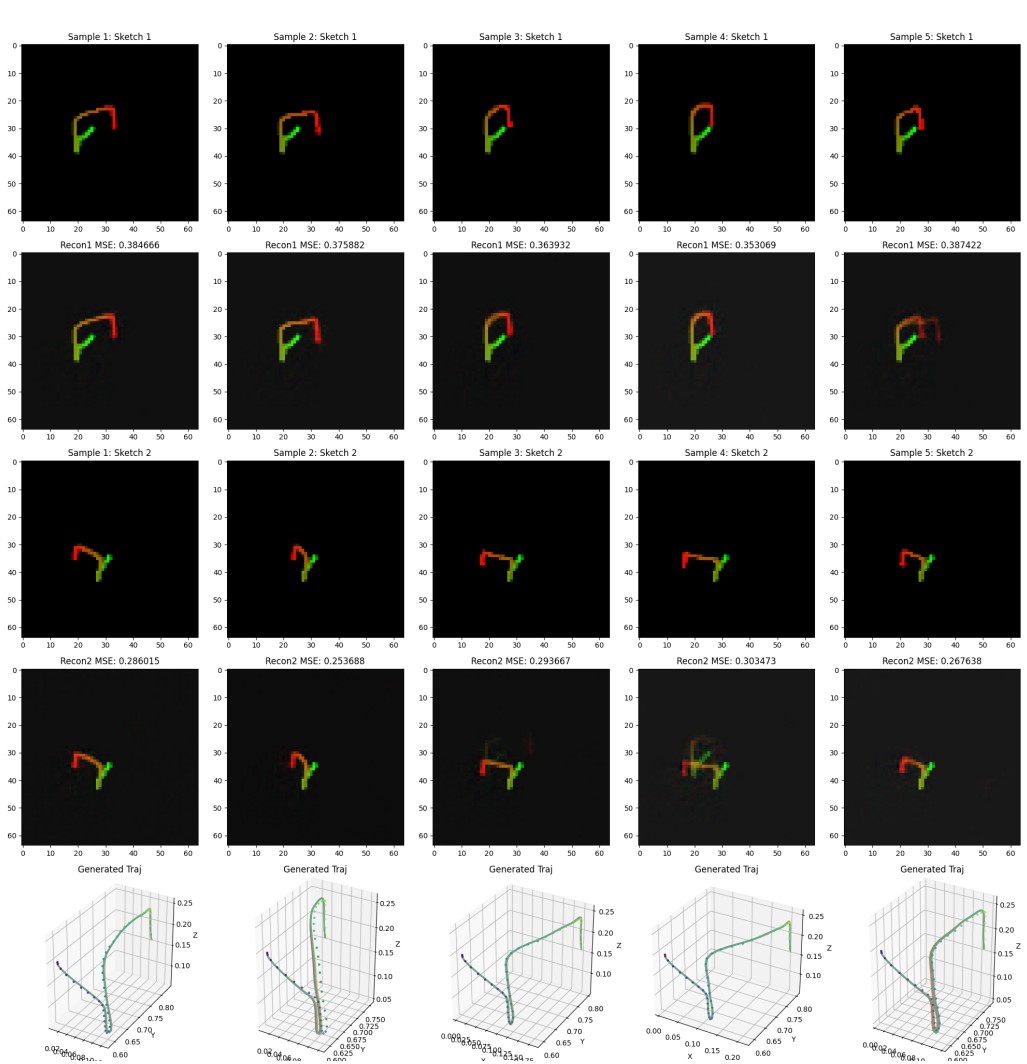

Figure 19: Trajectories with time-based color gradient for Assembly Metaworld simulation task.