# OpenReview forum: "Sketch-to-Skill: Bootstrapping Robot Learning with Human Drawn Trajectory Sketches"
_ICLR.cc/2025/Conference — Submitted to ICLR 2025_

### Official Review · Reviewer_cvP5 · 2024-10-30

**Soundness:** 1
**Presentation:** 3
**Contribution:** 1
**Rating:** 5
**Confidence:** 4

**Summary:**

This paper presents a method for training robotic manipulation policies through the use of 2D human trajectory sketches. The pipeline depends on a generator that takes a pair of 2D human trajectory sketches, drawn on the same scene captured from two different view points, and generates 3D trajectories (sequences of 3D points). This generator must be pre-trained using a dataset of sketches and corresponding 3D trajectories. Next, the 3D trajectories are used to train an agent with imitation learning, and then the agent is finetuned via reinforcement learning, using a discriminator-based reward to ensure consistency between the agent behavior and the 3D trajectories that came from the sketches. The method is shown to perform well on 6 tasks in MetaWorld and on a button pressing task in the real world.

**Strengths:**

The paper is well-written and easy to follow. The idea is interesting at a high-level and demonstrated on MetaWorld, a common benchmark, which could help others reproduce the work.

**Weaknesses:**

**Practicality of Approach.** The approach depends on a trained sketch-to-3D trajectory generator, which itself must be trained on a sufficient set of data. It seems that the burden of collecting this dataset would outweigh the benefits of using this generator as an alternative to providing human demonstrations. Furthermore, teleoperating a handful of demonstrations seems like a more general approach. Demonstrations can be collected for any task that the human can demonstrate, while the performance of the proposed approach is constrained to the capabilities of the trained generator. For example, it seems that the current method can only be using for position-controlled tasks, and not tasks where the robot must control both position and orientation. Furthermore, the trained generator might not generalize well beyond the training data, which likely needs to be collected on the same set (or very similar set) of tasks, or at the very least in the same domain (e.g. same robot, same workspace).

**Method Novelty.** The method itself seems limited in novelty -- the core component is the learned sketch-to-3D trajectory generator. Apart from that, it seems like an existing RL-finetuning algorithm is used (IBRL), with the only critical changes being (1) the choice of initial dataset and (2) the use of a discriminator. However, the discriminator itself seems to be unneeded, having little to no performance gain over not using it (Figure 8), while needed careful tuning for the discriminator reward weight (Figure 9). Even in the real robot experiments, it seems as though no discriminator is used.

**Experiment Issues.** The tasks presented in the paper appear to be limited in complexity. For example, the TD3 (RL baseline) can solve several of the tasks with just a sparse reward (Figure 5). I would suggest using some more challenging tasks for evaluation, perhaps from some other common benchmarks such as [robomimic](https://robomimic.github.io/), [RLBench](https://sites.google.com/view/rlbench), or [ManiSkill](https://www.maniskill.ai/home). There is also insufficient evaluation on some of the components of the method. In particular, how well does the sketch-to-3D generator generalize to new settings? Can it be applied to new tasks or new domains outside of the training data? How important is the architecture choice of the generator? Is the VAE crucial, or would other architectures work just as well? Is the choice of RL-finetuning method also critical, or would another method such as GAIL (as used in this work: https://arxiv.org/abs/1802.09564) work as well? Finally, in the real-world experiment, if BC already achieves 80%, what is the point of RL finetuning, if it achieves the same performance? Furthermore, training the sketch generator seemed to require 85 teleoperated trajectories itself (from Appendix). Training the 3D generator and then running RL seems much more painful than just training an agent directly on the teleoperated trajectories.

**Minor Issues.**

- line 71: "a more approach" - missing a word?

**Questions:**

- Sec 4.1 - is there any quantitative evaluation as well, or is it all qualitative?
- What if the two human sketches are inconsistent? It might be tricky to ensure that they are consistent.

---

> ### Author Response · Authors · 2024-11-24
> **Response to Reviewer 5 cvP5 (1)**
>
> We sincerely thank the reviewer for their thoughtful feedback and recognition of the paper's clear writing and successful demonstration on a common benchmark. We address each concern below.
> (Additional experiments and evaluation are reported in the appendix in the new version of the paper and marked in blue color).
>
> **Practicality of Approach**
>
> *R5: "The approach depends on a trained sketch-to-3D trajectory generator... the burden of collecting this dataset would outweigh the benefits..."*
>
> The initial data collection is actually quite efficient. We use "play data" that only needs to be collected once during robot setup. This data can come from:
>
> - Direct hardware interaction in a few hours;
> - Simulation, which requires no physical robot time;
> - Existing recorded trajectory datasets transformed to match the workspace coordinates;
> - A mix of these approaches.
>
> Once trained, the generator works across new tasks without retraining. This one-time investment is much more efficient than collecting new teleoperated demonstrations for every task. Our experiments demonstrate that the pretrained generator, trained on 24k trajectories, is sufficiently robust to support sketch-based learning across a variety of tasks.
>
> *R5: "The current method can only be used for position-controlled tasks..."*
>
> It's true that our sketches show just the path, but that doesn't mean our method can only be used for position-controlled tasks. These sketch trajectories act as a starting point, a way to guide exploration, while the RL component learns the complete task including orientation and gripper controls. We demonstrate this with additional experiments in RoboMimic’s Can Pick and Place task, our system successfully learns proper grasping angles and gripper actuation despite sketches only showing the reaching and placing paths. This showcases a key strength of our approach - we can bootstrap complex learning from simple 2D sketches while letting the RL discover the full spectrum of required controls. Think of the sketches as giving the robot a rough idea of "where to go," while letting it figure out the details of "how to get there" through interaction.
>
> **Method Novelty**
>
> *R5: "The method itself seems limited in novelty... the discriminator itself seems to be unneeded..."*
>
> While we build on existing components like IBRL, our key contribution is the novel integration of sketches into RL. This is fundamentally different from prior work that only used sketches for policy conditioning. We demonstrate that even approximate sketch-based demonstrations can effectively bootstrap RL. Regarding the discriminator - our new experiments on more complex tasks like Assembly (MetaWorld, Figure 18) show clear benefits: the success rate improves from 0.2 to 0.85 with the discriminator. Even in simpler tasks, the discriminator helps stabilize learning.
>
> **Expanded Task Complexity**
>
> *R5: "The tasks presented in the paper appear to be limited in complexity..."*
>
> To demonstrate our method's capabilities on more challenging tasks, we conducted extensive experiments on complex manipulation benchmarks:
>
> - **MetaWorld Assembly Task:** A two-stage task, classified as “hard” [1], where the robot first locates and reaches a peg, then picks it up and assembles it with a cylinder. In this task, our framework achieved a success rate of 93%, demonstrating its capability in handling precise manipulations under complex conditions.
> - **RoboMimic Can Pick and Place:** Another two-stage task, where the robot must first locate and reach a can, then pick it up and place it in a designated bin. Robomimic [2] is well established benchmark with demonstrations collected by human teleportation. The policies require predicting the orientation of the gripper as well and are longer than in MetaWorld. This task tested our policy's ability to predict the orientation of the gripper, with a success rate improving to 98% after refining our reinforcement learning approach.
> - **Real-World Tasks:** We tested our framework on two additional tasks, *Toaster Button Press* (single-stage) and *Bread Pick Place* (two-stage). These real-world environments included added clutter to simulate more realistic settings. The success rates here were 80% within 30k interactions steps and 95% within 12k interactions steps, respectively, showcasing the framework's adaptability to diverse and unstructured environments.

---

> ### Author Response · Authors · 2024-11-24
> **Response to Reviewer 5 cvP5 (2)**
>
> **Generator Component Analysis and Design Choice**
>
> *R5: "how well does the sketch-to-3D generator generalize to new settings? Can it be applied to new tasks or new domains outside of the training data? Is the architecture choice critical?”*
>
> The generalization capability of our sketch-to-3D generator is demonstrated through several experiments. In the latent space interpolation results (Figure 12), we show smooth transitions between different trajectories, indicating the model learns a continuous and meaningful representation that can generalize to unseen trajectories. Our generator is trained on randomly generated trajectories in simulation, and we demonstrate it works effectively across all the diverse tasks presented in the paper.
>
> Regarding architecture choices, our experiments with various models led us to the VAE-based design for several reasons. The VAE's latent space provides natural support for generating multiple plausible trajectories from a single sketch pair by adding controlled noise - a feature that proved valuable for policy learning. We tested alternatives including standard CNNs and MLPs, but found they either struggled with trajectory consistency across views or lacked the ability to generate diverse yet plausible trajectories.
>
> **Real-world BC vs RL performance**
>
> *R5: "in the real-world experiment, if BC already achieves 80%, what is the point of RL finetuning, if it achieves the same performance?"*
>
> While Behavior Cloning does indeed show high initial success, reaching up to an 80% success rate in our tests, the real advantage of RL fine-tuning comes into play under varied environmental conditions. RL fine-tuning is designed to enhance adaptability and robustness, qualities that are essential in practical deployments where environments cannot be perfectly controlled. This adaptability ensures that the robot can handle unexpected variations effectively, something BC might not consistently manage.
>
> **Efficiency and Justification for Data Use in Training the Generator**
>
> *R5: “Furthermore, training the sketch generator seemed to require 85 teleoperated trajectories… Training the 3D generator and then running RL seems much more painful …”*
>
> Initially, training our sketch-to-3D trajectory generator with 85 teleoperated trajectories might seem resource-intensive. However, this should be seen as a strategic, one-time investment that substantially enriches our training dataset, enabling the generator to produce versatile and adaptive responses without needing new data for each task.
>
> There are several ways to make this initial data collection even more efficient:
>
> - The data can be collected in simulation with matching camera setups, requiring no physical robot time;
> - We can utilize existing trajectory datasets by transforming them to match our workspace coordinates;
> - The collected data can be augmented to create a richer training set.
>
> Once trained, the generator serves as a powerful tool that translates simple sketches into useful demonstrations, eliminating the need for specialized hardware or expertise in data collection. This approach reduces the long-term burden of demonstration collection, increasing the scalability and efficiency of our deployment.
>
> **Application Across Multiple Real-World Experiments:** The utility and efficiency of our data use are exemplified by the ability to apply the trained generator to various tasks without retraining. For instance, once our generator was fine-tuned during the initial setup, it was adeptly applied to tasks like Toaster Button Press and Bread Pick Place (as shown in figure 16 and 17) in real-world environments with consistent camera setups. This adaptability underscores the practical value of our initial data investment and highlights the framework’s capability to generalize across different tasks effectively.

---

> ### Author Response · Authors · 2024-11-24
> **Response to Reviewer 5 cvP5 (3)**
>
> **Clarifications**
>
> *R5: ” Sec 4.1 - is there any quantitative evaluation as well, or is it all qualitative?” “What if the two human sketches are inconsistent? It might be tricky to ensure that they are consistent.”*
>
> While we don't have quantitative comparisons of generator performance under different sketch variations, our design focuses on building in robustness through two complementary approaches. During training, we apply specific augmentation strategies:
>
> - First, we employ image augmentations that only update the VAE component, including random rotation (±25°), random translation (±20% of image size), scaling variations (0.8-1.5x), and Gaussian noise (σ = 0.01). This helps the model learn robust sketch representations.
> - Second, we apply controlled elastic deformations to both sketches and trajectories when training the full generator. The sketches undergo elastic deformation with alpha=50.0 and sigma=5.0, while trajectories are perturbed with Gaussian noise (σ = 0.015) before refitting and resampling to ensure smoothness. This second strategy specifically addresses potential inconsistencies in hand-drawn sketches.
>
> These carefully designed augmentations simulate the natural variations in human-drawn sketches. The effectiveness of this approach is demonstrated qualitatively through our latent space interpolation results (Figure 12), where we show the generator produces smooth, coherent trajectories even between quite different sketch pairs.
>
> **Minor Points**
>
> We have addressed the typographical error on line 71 and other minor issues throughout the document.
>
> [1] Seo, Y., Hafner, D., Liu, H., Liu, F., James, S., Lee, K., & Abbeel, P. (2023, March). Masked world models for visual control. In *Conference on Robot Learning* (pp. 1332-1344). PMLR.
>
> [2] Mandlekar, Ajay, et al. "What matters in learning from offline human demonstrations for robot manipulation." *arXiv preprint arXiv:2108.03298* (2021).

---

> > ### Comment · Reviewer_cvP5 · 2024-11-26
> > **Response**
> >
> > Thanks for the thorough response and additional experiments - I greatly appreciate it.
> >
> > > Method Novelty
> >
> > I appreciate the additional experiment showing the value of the discriminator, but I still feel that the novelty of the approach is still limited.
> >
> > > Generator Component Analysis and Design Choice
> >
> > It would have been nice to see some quantitative comparison of these different choices if possible, to inform others on how critical the choice of VAE is.
> >
> > I am still leaning towards rejection, but I have raised my score, since I believe the additional experiments provided have improved the quality of the work.

---

> > > ### Author Response · Authors · 2024-11-27
> > > **Response to Reviewer 5 cvP5 (4)**
> > >
> > > Thank you for your feedback and for acknowledging the improvements in our paper. We appreciate your adjusted score. In response to your comments on the generator component analysis, we've added new results in Appendix E4.
> > >
> > > (Additional experiments and evaluation are reported in the appendix in the new version of the paper and marked in red color).
> > >
> > > *R5: “some quantitative comparison of these different choices if possible…”*
> > >
> > > We conducted an ablation study comparing using VAE versus replacing the VAE with a simple CNN (with the same architecture as the VAE's encoder, minus the decoder and loss components) based network. This experiment involved 1000 trajectories with an 80:20 validation split. The results, detailed in Table 2, demonstrate that the VAE (Figure 2), significantly improves performance across key metrics including reconstruction loss and trajectory loss, showcasing how critical the choice of VAE is.
> > >
> > > We hope these additions address your concerns and provide a clear basis for the design choices in our method. Thank you for your insights, and we are open to further discussion to refine our work.

---

> > > > ### Author Response · Authors · 2024-12-02
> > > > **Follow-Up on Revised Submission and Additional Experiments : Reviewer 5 cvP5 (5)**
> > > >
> > > > Dear Reviewer cvP5,
> > > >
> > > > I hope this message finds you well. We wanted to express our gratitude once again for the constructive feedback you provided. We have updated the manuscript with additional experiments in Appendix E4, as suggested in your comments, to further substantiate the design choices and effectiveness of the VAE in our framework.
> > > >
> > > > As the review process is nearing its conclusion, we would appreciate any further comments or questions you might have regarding the updates we've made. Additionally, if you believe that the revisions and additions have adequately addressed your concerns, we kindly ask you to consider whether an increase in your review score might be warranted.
> > > >
> > > > Thank you once again for your time and consideration. We look forward to your response.

---

### Official Review · Reviewer_ne7n · 2024-11-02

**Soundness:** 3
**Presentation:** 4
**Contribution:** 3
**Rating:** 8
**Confidence:** 4

**Summary:**

This work presents an interesting approach to extending IBRL by incorporating a trajectory generation method reminiscent of RT-Sketch. The core idea of generating 3D trajectories from pairs of 2D sketches, augmented by noise injection for data diversity, is novel.  The addition of a discriminator to guide the IBRL policy towards these generated trajectories further strengthens the link between user-provided sketches and learned robot behavior. However, the paper leaves some questions unanswered.

Firstly, the authors show a pair of sketches in Fig. 2, but it's not quantitatively clear how robust the trajectory generator is to variations in the spatial correlation between these sketches. Some of the analysis is present in appendix B, but adding information like the benefits of each augmentation strategy, sample variations in sketches, etc. would be beneficial. For example, how much spatial variation do you add?

In line 369, how many hand-drawn sketches do you collect?. Is it also three (i.e. number of expert tele-operated policies used)? The ablation study on Discriminator reward weighting in Fig. 9 is good, but the choice of reward weights seems arbitrary. A more logical progression (e.g., 0.1, 0.01, 0.001) would make the analysis clearer, or explain your rationale for choosing the current set of weights. While the appendix includes a no-discriminator condition (Fig 14), this crucial ablation should be in the main paper.

Looking ahead, the authors briefly mention "time parameterization of the trajectory" in future work (line 529).  This raises the question of how they envision solving the increased difficulty in achieving correlation between two sketches when time is factored in.

Finally, the paper would benefit from a discussion situating this method within the broader context of recently popular Vision-Language Agents (VLAs) for robotic manipulation, such as OpenVLA. This is particularly relevant given the increasing use of real-world robot-collected data in VLAs, whereas Sketch-to-Skill is trying to reduce tele-operated data collection. How do the authors see sketches complementing this data? Moreover, given the prevalence of LFD and imitation learning in VLA pre-training and fine-tuning, how do the authors position the advantages of RL in this setting? Including this in a related work or discussion section would be helpful.

Some minor points:
*   There is a typo on line 071: "we present a more approach."
*   For improved accessibility, the figures could be made more color-blind friendly, potentially by incorporating textures. For example, add textures to the bar plots in Fig 7.
*   In Fig. 3, why is there an asterisk after "Replay Buffer"?

**Strengths:**

provided in summary

**Weaknesses:**

provided in summary

**Questions:**

provided in summary

---

> ### Author Response · Authors · 2024-11-24
> **Response to Reviewer 4 ne7n**
>
> We sincerely appreciate the reviewer's thorough assessment and recognition of the novelty in our approach. We are grateful for the insightful questions and suggestions, which have helped us strengthen our work. Below, we address each point and provide additional information to clarify the concerns raised. (Additional experiments are reported in the appendix in the updated paper, marked in blue).
>
> **Trajectory Generator Robustness to Spatial Correlation Variations**
>
> *R4: "It's not quantitatively clear how robust the trajectory generator is to variations in the spatial correlation between these sketches."*
>
> While we don't have quantitative comparisons of generator performance under different sketch variations, our design focuses on building in robustness through two complementary approaches. During training, we apply specific augmentation strategies:
>
> - First, we employ image augmentations that only update the VAE component, including random rotation (±25°), random translation (±20% of image size), scaling variations (0.8-1.5x), and Gaussian noise (σ = 0.01). This helps the model learn robust sketch representations.
> - Second, we apply controlled elastic deformations to both sketches and trajectories when training the full generator. The sketches undergo elastic deformation with alpha=50.0 and sigma=5.0, while trajectories are perturbed with Gaussian noise (σ = 0.015) before refitting and resampling to ensure smoothness. This second strategy specifically addresses potential inconsistencies in hand-drawn sketches.
>
> These carefully designed augmentations simulate the natural variations in human-drawn sketches. The effectiveness of this approach is demonstrated qualitatively through our latent space interpolation results (Figure 12), where we show the generator produces smooth, coherent trajectories even between quite different sketch pairs.
>
> **Number of Hand-Drawn Sketches Collected**
>
> *R4: "In line 369, how many hand-drawn sketches do you collect? Is it also three (i.e. number of expert tele-operated policies used)? "*
>
> Yes, for each task, we collected three hand-drawn sketches, corresponding to the number of expert tele-operated demonstrations used in our experiments. We have clarified this in the main text (line 369) to ensure consistency and avoid confusion.
>
> **Reward Weighting in Discriminator Ablation**
>
> *R4: "The choice of reward weights seems arbitrary. A more logical progression (e.g., 0.1, 0.01, 0.001) would make the analysis clearer."*
>
> Thank you for your feedback regarding the choice of reward weights in our ablation study. In response to your comment, we selected reward weights (0.1, 0.5, 0.05, 0.005) for our experiments on the MetaWorld task 'CoffeePush', which are presented in Figure 9 of our paper.
>
> **Future Work on Time Parameterization**
>
> *R4: "The authors briefly mention "time parameterization of the trajectory" in future work (line 529).This raises the question of how they envision solving the increased difficulty in achieving correlation between two sketches when time is factored in."*
>
> We envision solving time correlation challenges by incorporating visual cues like color gradients in sketches to represent temporal information.
>
> **Relation to Vision-Language Agents (VLAs)**
>
> *R4: "The paper would benefit from a discussion situating this method within the broader context of recently popular Vision-Language Agents (VLAs) for robotic manipulation."*
>
> Thank you for this insightful suggestion about positioning our work in the context of VLAs. We see sketches as complementary to the language-based instruction paradigm of VLAs in several ways:
>
> - While VLAs excel at understanding high-level task descriptions, sketches provide intuitive spatial guidance that can be challenging to convey through language alone;
> - Sketches could serve as an additional modality for VLAs, particularly for specifying precise trajectories or motion patterns that are difficult to describe verbally;
> - Our sketch-based demonstrations could potentially enhance VLA pre-training by providing additional structure to exploration.
>
> Regarding the choice of RL versus pure imitation learning in this context, RL offers key advantages:
>
> - It allows the policy to discover better solutions than the sketch-generated demonstrations while using them as a starting point;
> - The policy can adapt to variations not captured in the sketches, like precise object positions or orientations;
> - The discriminator-guided exploration helps maintain consistency with human intent while optimizing for task success.
>
> **Minor Points**
>
> - We have corrected the typo on line 071.
> - To improve accessibility, we have updated the figures to be more color-blind friendly. We have changed the colors of the Figure 5 and 6, and incorporated textures in the bar plots of Figure 7.
> - The asterisk after "Replay Buffer" in Fig. 3 indicates that the buffer is initialized with the open-loop servoing demonstrations. We have added a clarifying note in the figure caption.

---

### Official Review · Reviewer_beAX · 2024-11-02

**Soundness:** 2
**Presentation:** 3
**Contribution:** 3
**Rating:** 8
**Confidence:** 4

**Summary:**

This work aims to use human-drawn sketches to learn RL policies for robotic manipulation tasks. They use a trajectory generator to generate demonstrations from sketches, then use IBRL to train a policy from the generated demonstrations. The paper additionally tests whether there are benefits in (1) generating more demonstrations per sketch and (2) performs guided exploration using a discriminator.

**Strengths:**

Good results on a novel method, which is able to generally preserve performance while removing the need for high-quality teleoperated demonstration data. Each piece of the method is explained clearly, and the method is designed logically (e.g. exploit the demonstrated smoothness of the latent space to generate more demos per sketch with controlled noise). The results are presented clearly and effectively, and the Appendix provides useful information. Tests in the real-world show practical applicability of the method.

**Weaknesses:**

Firstly, some additions to improve performance seem to have ambiguous results:

(a) Discriminator to guide exploration: In Fig. 6, the performance with and without the discriminator seem fairly close. The policy trained with discriminator only seems to perform better in ButtonPressTopdownWall, while the policy without discriminator seems to perform the same if not marginally better in the rest of the tasks.

(b) Additional demos per sketch: In Fig. 8, 10 demos per sketch + discriminator has similar average performance in ButtonPress (though lower lows), similar performance to others in CoffeePush, and has better average but lower max performance in BoxClose. It is mentioned that there is diminishing performance with more demos, but it does not seem clear that more demos per sketch is necessarily better? In fact, in Fig. 14, 10 demos + discriminator seems worse than others.

Secondly, it seems one benefit of sketching vs teleoperating is that sketched demonstrations can be made much faster and without expensive hardware. So, the usage of only 3 demos, and the absence of demo ablations (specifically on the number of sketches/teleoperated demonstrations provided) is somewhat odd. Perhaps MetaWorld tasks are too easy when using more demonstrations, but there exist harder standard LfD benchmarks which the authors did not test on.

**Questions:**

Are there additional clarifications which would explain why the authors chose to keep the discriminator and 10 demos per sketch? Perhaps a different way to present the data (e.g. raw success rate on more eval episodes)?

Does the method scale to more sketches (and how does it compare to IBRL when providing more teleoperated data)?

Have you tried this method with other backbone RL algorithms (e.g. ensemble SAC, REDQ) to improve sample efficiency or final performance?

---

> ### Author Response · Authors · 2024-11-24
> **Response to Reviewer 3 beAX (1)**
>
> We sincerely thank the reviewer for their thoughtful feedback and recognition of the strengths in our work, including its novel approach, logical design, and practical applicability. Below, we address the concerns and provide clarifications, incorporating additional results to strengthen our contributions. (Additional experiments and evaluation are reported in the appendix in the new version of the paper and marked in blue color).
>
> **Discriminator Contribution:**
>
> *R3: "In Fig. 6, performance with and without discriminator seem fairly close..."*
>
> We agree the discriminator's impact varies across tasks. While the overall performance appears similar in Figure 6, the discriminator's value becomes clear when examining different task complexities. We conducted additional experiments on more challenging environments to demonstrate the benefit of using discriminator - specifically Assembly (a MetaWorld [1] hard task, Figure 18) and Pick and Place Can (from RoboMimic [2], which offers significantly higher difficulty than MetaWorld, Figure 15). In these complex manipulation tasks, as well as in ButtonPressTopdownWall (Figure 5, 6), the discriminator significantly improves performance by helping the policy better align with sketch-generated trajectories. For instance, in ButtonPressTopdownWall, it improves final success rate from 0.4 to 0.75.
>
> For simpler tasks like ButtonPress and Reach, both versions achieve similar final performance (~0.95 success rate). This makes sense - in straightforward tasks, standard exploration strategies already work well. However, we kept the discriminator as a standard component since it ensures reliable performance across all scenarios, especially those requiring precise control, without harming performance in simpler cases.
>
> **Number of Generated Demonstrations per Sketch:**
>
> *R3: "It is not clear that more demos per sketch is necessarily better... In fact, in Fig. 14, 10 demos + discriminator seems worse than others.”*
>
> You raise an excellent point about the varying impact of demonstration numbers across different tasks. Looking at Figures 8 and 14 more closely:
>
> For simple tasks like ButtonPress, Reach and ReachWall, we observe that:
>
> - Even with slightly worse performance using 10 demos, the difference is minimal
> - 1-3 demos per sketch are sufficient for these tasks, as they involve straightforward point-to-point movements
> - Additional demonstrations provide little benefit since basic exploration can effectively solve these tasks
>
> However, for more complex tasks, like BoxClose and ButtonPressTopdownWall:
>
> - 10 demos show clear benefits in reducing performance variability and improving stability
> - These tasks require more precise control and have more failure modes, making the broader coverage from additional demonstrations valuable
> - The extra demonstrations help create a more comprehensive experience dataset that better captures task complexity
>
> This task-dependent pattern suggests that while simpler tasks can be effectively learned from fewer demonstrations, more complex manipulation tasks benefit from the richer training signal provided by additional demonstrations generated per sketch. This is one of the strengths of our approach. We can generate more demos but from same sketches, essentially not increasing the effort on the part of the user

---

> > ### Author Response · Authors · 2024-11-24
> > **Response to Reviewer 3 beAX (2)**
> >
> > **Scaling with More Sketches**
> >
> > *R3: "The usage of only 3 demos, and absence of demo ablations is somewhat odd... sketched demonstrations can be made much faster and without expensive hardware."*
> >
> > Thank you for highlighting both the potential limitations and key benefits of our approach - particularly the speed of sketch collection and elimination of specialized hardware requirements. You make a valid point about demo scaling.
> >
> > The choice of three demonstrations was deliberate, aiming to demonstrate our framework's effectiveness under minimal data conditions - a crucial consideration for real-world applications where extensive demonstration collection is impractical. In MetaWorld tasks (Figures 5-6), three sketches achieve 96% of teleoperated demonstration performance while being significantly faster to collect and requiring no specialized hardware.
> >
> > To validate scalability to more complex tasks, we tested on RoboMimic's Can Pick and Place task and MetaWorld's hard Assembly task. These benchmarks are more challenging than the initial MetaWorld tasks:
> >
> > - Both tasks demand accurate multi-step manipulations, requiring adaptation to variable object positions and precise grasping
> > - We used 3 sketches for the Assembly task to keep it consistent with the rest of MetaWorld tasks, while the RoboMimic Can Pick and Place task being harder required 20 sketches
> >
> > The success on these more challenging benchmarks demonstrates our method effectively scales when task complexity demands more demonstrations.
> >
> > **Alternative RL Algorithms**
> >
> > *R3: "Have you tried this method with other backbone RL algorithms (e.g. ensemble SAC, REDQ) to improve sample efficiency or final performance?"*
> >
> > We have not tested other backbone RL algorithms. We chose TD3 as our backbone algorithm for its established stability and performance in continuous control tasks. Our framework is algorithm-agnostic - the core contribution lies in using sketches to bootstrap and guide policy learning, which could be integrated with other RL algorithms. TD3 serves as an effective baseline for demonstrating this contribution.
> >
> > Additionally, since our method already achieves 96% of the teleoperated demonstration performance, the choice of base RL algorithm is less critical than the demonstration bootstrapping and guidance mechanism we introduce.
> >
> > [1] Yu, Tianhe, et al. "Meta-world: A benchmark and evaluation for multi-task and meta reinforcement learning." *Conference on robot learning*. PMLR, 2020.
> >
> > [2] Mandlekar, Ajay, et al. "What matters in learning from offline human demonstrations for robot manipulation." *arXiv preprint arXiv:2108.03298* (2021).

---

> > > ### Comment · Reviewer_beAX · 2024-11-26
> > >
> > > Thank you for your response, my questions regarding the discriminator, number of generated demonstrations, and alternative RL algorithms have been addressed. However, I still find the practical applicability of the method questionable, since such few demos are used for training.
> > >
> > > The main drawback of teleoperation is that it is time-consuming to collect many demonstrations; sketches could alleviate this, and drawing 100s of feasible sketches would be quite straightforward. If only 3-10 teleoperated demonstrations are needed, then I find little need to use the Sketch-to-Skill method (which has slightly degraded performance) compared to collecting a few regular demonstrations.
> > >
> > > That being said, while I believe there is room for improvement in the evaluation (more difficult tasks + scaling), I view the generator with smooth and controllable latent space as an important contribution, and the method itself is reasonable. I also acknowledge the question of scaling and more difficult tasks is something that can be tackled in future work, and would be difficult to incorporate on top of the ablations and real hardware experiments.
> > >
> > > Seeing as most of my concerns are addressed, I have decided to increase my score to 8, accept.

---

> > > > ### Author Response · Authors · 2024-11-27
> > > > **Response to Reviewer 3 beAx (3)**
> > > >
> > > > Thank you for your thoughtful review and for increasing your score to an accept. We greatly appreciate your recognition of the work we've put into addressing your initial concerns. Thank you again for your constructive feedback, it helped us tremendously in improving our work!

---

### Official Review · Reviewer_k4pp · 2024-11-03

**Soundness:** 3
**Presentation:** 3
**Contribution:** 1
**Rating:** 3
**Confidence:** 4

**Summary:**

This paper explores the use of a sketch-based method as the means to provide robots with expert demonstrations that show how various manipulation tasks should be completed. These demonstrations can then be used to bootstrap the process of training control policies via behaviour cloning and reinforcement learning. The proposed system consequently consists of 1) a module that transforms a pair of 2d sketches (which correspond to two different viewpoints of a scene) into 3d trajectories, 2) a behaviour cloning step to generate an initial control policy, and 3) an RL module that refines this policy. Comparisons to teleoperation-based demonstrations and pure RL show how well this method works for six manipulation tasks that are relatively simple.

**Strengths:**

I like the way the paper is written, and the clear research questions that are articulated / studied in Chapter 4. Within the context of the relatively simple manipulation tasks which are chosen, I think the paper is very thorough with its analysis, and the evaluation does indeed shine a positive light on sketch-based methods as the means to provide expert demonstrations. Nevertheless, as I explain below, I am afraid that this approach will not scale to manipulation tasks which are more complex.

**Weaknesses:**

The methodology is overall sound, but nothing stands out as particularly novel. In addition, and more importantly, the manipulation tasks which are chosen for experiments are too basic. They do not showcase adaptive behaviours (e.g. what happens if an object that needs to be picked up is accidentally dropped, or if it otherwise does not behave/move as originally intended) or multi-stage, longer-horizon motions (e.g. opening a cardboard box before picking objects up from it, or folding a t-shirt). Teleoperation has been shown to be effective in such settings due to its real-time feedback and high-throughput high-DOF nature. I fear that the sketch-based approach presented here is simply too limited and will quickly become overly cumbersome as the task difficulty increases. For example, even providing a time-dependent orientation for the robot's end effector would likely be non-trivial through the pair of 2d sketches used as input for the system that is proposed in this paper. I would be happy to reconsider this standpoint if further experiments are presented for manipulation tasks that are considerably more complex.

**Questions:**

As mentioned above, I have significant concerns over the scalability of the proposed sketch-based method for complex, real-world manipulation tasks. As robotics manipulation problems go, the tasks which are chosen in this work fall well within the toy-model domain, and as such, the findings do not apply to the real-world problems which we expect robots to be able to learn how to undertake through expert demonstrations.

---

> ### Author Response · Authors · 2024-11-24
> **Response to Reviewer 2 k4pp (1)**
>
> We sincerely appreciate the reviewer's positive feedback on the clarity of our paper and the thoroughness of our analysis within the chosen manipulation tasks. We also thank the reviewer for raising important concerns about the scalability and complexity of our approach. Below, we address these concerns and provide additional insights and results to strengthen our contributions. (Additional experiments and evaluation are reported in the appendix in the new version of the paper and marked in blue color).
>
> **1. Role and Scope of Sketch-Based Demonstrations**
>
> Our framework is designed to use sketch-based demonstrations as an accessible and low-cost method to initialize policy learning, particularly in scenarios where teleoperation is infeasible or prohibitively expensive. We do not position sketches as a complete replacement for high-fidelity demonstrations but rather as a practical supplement in data-scarce or constrained environments. This distinction underscores the intended utility of our approach in complementing existing methods.
>
> **2. Addressing Task Complexity with Expanded Experiments**
>
> *R1: "I fear that the sketch-based approach presented here is simply too limited and will quickly become overly cumbersome as the task difficulty increases."*
>
> We acknowledge the reviewer's valid concern about the scalability of our method to more complex, real-world manipulation tasks. To address this, we have expanded our experimental evaluation to include more challenging and dynamic scenarios:
>
> - **Additional Real-world Experiments: Toaster Button Pressing** and **Bread Pick-and-Place.** These tasks involve interacting with cluttered, real-world environments, requiring precise planning and execution. The results, documented in Figures 15 and 16, illustrate the scalability of our method beyond basic scenarios.
> - **Benchmarks with Long-Horizon, Multi-Stage Manipulations**: We conducted experiments on challenging tasks such as **Pick and Place Can** (from RoboMimic) and **Nut Assembly** (from  MetaWorld). These tasks require sequential reasoning and adaptation, highlighting our framework’s applicability to longer and more intricate task horizons. Detailed results and comparisons are included in the appendix.
>
> Through these expanded experiments, our approach demonstrates robustness and adaptability, addressing concerns about its scalability to real-world applications.
>
> **3. Addressing Challenges of Time-Dependent and Orientation-Specific Tasks**
>
> *R2: "Even providing a time-dependent orientation for the robot's end effector would likely be non-trivial through the pair of 2d sketches used as input for the system that is proposed in this paper.”*
>
> We acknowledge the reviewer’s concerns regarding the difficulty of encoding time-dependent or orientation-specific information in sketches. While these challenges are not fully addressed in the current scope, we view them as natural extensions of our approach. For instance:
>
> - Incorporating **visual markers or gradients** within sketches to encode temporal and orientation-specific information, as inspired by research like the RT Trajectory paper, is a promising direction.
> - In tasks like the **RoboMimic Pick and Place Can**, the sketch-based demonstrations provide initial position cues, while our framework leverages reinforcement learning (RL) to refine the policies, dynamically adjusting for orientation and time-specific factors during execution. This approach does not require fully detailed trajectory information, as the sketches primarily guide exploration during RL. Notably, our results are comparable to IBRL, which relies on human demonstrations with explicit orientation details (Figure 15). This highlights the effectiveness and robustness of our sketch-based method in handling orientation and other task complexities without needing fully specified inputs.

---

> > ### Author Response · Authors · 2024-11-24
> > **Response to Reviewer 2 k4pp (2)**
> >
> > **4. Sketch-Based Demonstrations vs. Teleoperation**
> >
> > *R2: "Teleoperation has been shown to be effective in such settings due to its real-time feedback and high-throughput high-DOF nature."*
> >
> > While teleoperation excels in tasks requiring real-time feedback and high degrees of freedom, it comes with practical limitations such as latency and the need for expert operators. Sketch-based methods offer complementary advantages:
> >
> > - **Scalability**: Sketches are easy to create and deploy, making them an effective tool for initializing policy learning across a variety of environments. The process of generating sketches is highly flexible: they can be created through human drawings, video demonstrations, or even by using advanced techniques such as prompting large language models (LLMs) with code for policies, image-generating models, or waypoint generation models. Furthermore, sketches do not require expertise from the robot operators themselves, enabling the possibility of crowdsourcing sketch generation via platforms like MTurk. This contrasts with teleoperation, which relies on skilled operators and is less scalable. As a result, sketches offer an accessible and scalable way to begin training policies, even in diverse or resource-constrained settings.
> > - **Adaptability**: Our experiments, particularly in tasks like Bread Pick-and-Place, demonstrate that sketch-guided policies can effectively handle dynamic interactions and unexpected conditions during execution. In these experiments, the environment was randomized with varying levels of clutter, using different objects of daily use in each episode. This introduced significant variability in the task, such as changes in initial gripper position, and interactions with other items. Despite this, our framework successfully adapted to these dynamic conditions, allowing the robot to reliably execute the task even when faced with new and unpredictable challenges
> >
> > By leveraging reinforcement learning and discriminator feedback, our framework builds on the initial guidance provided by sketches to adapt to real-world challenges, making it a scalable alternative or complement to teleoperation.
> >
> > **Conclusion**
> >
> > We appreciate the opportunity to address your concerns and clarify the scope and strengths of our approach. By extending our experiments to include more complex tasks and addressing the challenges raised, we have demonstrated that sketch-based methods can scale beyond simple scenarios to tackle realistic robotic manipulation tasks. Your feedback has been invaluable, and we look forward to further improving and refining our framework to meet these challenges.

---

> > > ### Author Response · Authors · 2024-11-27
> > > **Response to Reviewer 2 k4pp (3)**
> > >
> > > We hope our response has addressed your concerns. We would really appreciate it if you could consider updating your rating based on the clarification provided. Let us know if there’s anything else we can answer or clarify. Thank you for your valuable feedback!

---

> > ### Comment · Reviewer_k4pp · 2024-12-02
> > **reviewer feedback**
> >
> > Thank you for the detailed response. I appreciate the discussion and additional experiments, but I still believe strongly that we have not seen enough evidence to suggest that sketch-based demonstrations will be useful for real-world, in-the-wild tasks. Without controlling the end effector orientation to precisely show how a robot should undertake a particular task, one cannot guarantee that the resulting policy will be successful. Again, I am referring to the types of complex tasks, like folding clothes, where learning-by-demonstration is one of the best tools roboticist have at the moment. If sketch-based modelling is just a cheaper / more accessible alternative to teleoperation for simpler tasks, then the real-world impact of such a method will be limited. As such, I will maintain my original score.

---

> ### Author Response · Authors · 2024-12-02
> **Clarification on the Role of Sketch-Based Demonstrations**
>
> Thank you for your feedback. We agree with you that there are certain tasks where teleoperation is better suited for giving demonstrations. However, as evidenced by a number of recent papers investigating the use of sketches in robot learning, sketches are useful in practice. [1-5] The goal of sketches is to complement, rather than replace, high-fidelity demonstration techniques such as teleoperation. Sketches make robot learning more accessible and when high precision is not paramount or where teleoperation is costly or not feasible. As shown in prior research, even basic sketches can effectively initiate and guide the learning process for a variety of tasks (refer to [1-5]).
>
> Our approach using sketches is part of an evolving exploration within the robotics community. Sketch-based methods have been previously studied as high-level guidance for initiating robot learning, notably in tasks where exact precision is not the primary requirement. Our work builds on this foundation and extends its application to reinforcement learning (RL), showcasing how sketches can effectively guide policy development even without detailing every aspect of a task's execution.
>
> For example, in the RoboMimic Can Pick and Place task, our framework successfully learned necessary orientation adjustments through RL interactions, despite these details not being explicitly depicted in the sketches.
>
> Looking ahead, there is significant potential to enhance the utility of sketches in robotic instruction. Incorporating elements such as orientation markers or color-coded actions within sketches could provide more detailed control cues, broadening the scope of tasks to which this method can be effectively applied.
>
> [1] Zhi, Weiming, Tianyi Zhang, and Matthew Johnson-Roberson. "Instructing robots by sketching: Learning from demonstration via probabilistic diagrammatic teaching." 2024 IEEE International Conference on Robotics and Automation (ICRA). IEEE, 2024.
>
> [2] Gu, Jiayuan, et al. "RT-Trajectory: Robotic Task Generalization via Hindsight Trajectory Sketches." The Twelfth International Conference on Learning Representations.
>
> [3] Sakamoto, Daisuke, et al. "Sketch and run: a stroke-based interface for home robots." Proceedings of the SIGCHI conference on human factors in computing systems. 2009.
>
> [4] Ahmad, Haseeb, et al. "A sketch is worth a thousand navigational instructions." Autonomous Robots 45.2 (2021): 313-333.
>
> [5] Skubic, Marjorie, et al. "Using a hand-drawn sketch to control a team of robots." Autonomous Robots 22 (2007): 399-410.

---

### Official Review · Reviewer_vobB · 2024-11-04

**Soundness:** 2
**Presentation:** 2
**Contribution:** 2
**Rating:** 5
**Confidence:** 3

**Summary:**

SKETCH-TO-SKILL leverages 2D human-drawn sketches to bootstrap and guide RL for robotic manipulation, making the approach accessible and potentially impactful. However, based on the presented experiments, the practical potential remains unclear. The real-world demonstrations focus on relatively simple tasks, making it challenging to fully gauge the framework's effectiveness or scalability in more complex, nuanced environments. More diverse and demanding experiments, along with detailed explanations, would help reveal whether this approach can truly generalize and perform under varied real-world conditions.

**Strengths:**

- The paper introduces an innovative approach by leveraging human-drawn 2D sketches to initialize and guide reinforcement learning in robotic manipulation.

- SKETCH-TO-SKILL provides an alternative in robot learning by enabling task training from simple sketches, which reduces the reliance on teleoperation data, specialized hardware, and advanced expertise. This approach potentially broadens access to robotic training methods.

**Weaknesses:**

1. The examples shown focus on short, simple trajectories, so it’s difficult to gauge how well the framework would handle longer, more complex paths. This leaves open the question of how trajectory complexity and length impact the learning cost.

2. For human users, sketching complex trajectories might require a lot more effort than the straightforward examples presented.

3. There’s also limited information on how the framework would deal with cases where paths overlap or are partially obscured. This could present practical challenges, both in terms of creating the sketches and the model’s ability to interpret them accurately.

4. The paper doesn’t clarify if there are specific guidelines for choosing sketch perspectives. It seems each task might need its own camera setup, which could complicate consistency and generalization in real applications.

5. The discriminator doesn’t appear to contribute significantly to the results, which makes it unclear how essential it is or if it’s been optimized effectively for generalization.

6. The experiments are carried out in simplified environments without much complexity or occlusion, so it’s hard to assess how well the model would perform in more realistic, cluttered settings.

7. The real-world tasks demonstrated are quite basic, which doesn’t fully showcase the framework’s potential across a broader range of applications. More varied real-world tests could provide deeper insights into its versatility.

8. There’s little information on the types of sketches and common sketching errors anticipated. Understanding how the model handles imperfections and what types of errors might impact learning outcomes would add clarity.

**Questions:**

1. How does trajectory complexity affect the learning cost, both for the model and for users creating these sketches?

2. How feasible is it for users to sketch accurately for more complex, multi-step tasks, and is this approach realistic for those scenarios?

3. How does the model manage paths that overlap or are partially obscured? Are there any specific recommendations for users to handle these situations in sketches?

4. Are there guidelines for choosing the best perspective for sketching, and does each task require a different camera setup? If so, how might this impact the model’s ability to generalize?

5. Has the discriminator been optimized for generalization, and what tangible impact does it have on the model’s overall performance?

6. How sensitive is the model to changes in the scene, such as additional objects or occlusions?

7. Would testing on a wider variety of real-world tasks help to bring out a clearer picture of the model’s strengths and limitations?

8. What are the most common sketching errors expected in real-world use? How does the model handle these, and which types of errors are most likely to affect performance?

---

> ### Author Response · Authors · 2024-11-24
> **Response to Reviewer 1 vobB (1)**
>
> We sincerely thank the reviewer for their thoughtful feedback and recognition of SKETCH-TO-SKILL's innovative approach and potential for making robot learning more accessible through simple sketches. Below, we address the concerns about practical applicability and experimental validation, incorporating additional details and clarifications to strengthen our contributions. (Additional experiments and evaluation are reported in the appendix in the new version of the paper and marked in blue color).
>
> **Real-World Performance & Complex Task Handling:**
>
> *R1: "The examples shown focus on short, simple trajectories... experiments are carried out in simplified environments without much complexity or occlusion... real-world tasks demonstrated are quite basic..."*
>
> We have rigorously evaluated our framework's capability with complex trajectories and realistic environments through extensive additional experiments:
>
> - **MetaWorld Assembly Task:** A two-stage task, classified as “hard” [1], where the robot first locates and reaches a peg, then picks it up and assembles it with a cylinder. In this task, our framework achieved a success rate of 93%, demonstrating its capability in handling precise manipulations under complex conditions.
> - **RoboMimic Can Pick and Place:** Another two-stage task, where the robot must first locate and reach a can, then pick it up and place it in a designated bin. Robomimic [2] is well established benchmark with demonstrations collected by human teleportation. The policies require predicting the orientation of the gripper as well and are longer than in MetaWorld. This task tested our policy's ability to predict the orientation of the gripper, with a success rate improving to 98% after refining our reinforcement learning approach.
> - **Real-World Tasks:** We tested our framework on two additional tasks, *Toaster Button Press* (single-stage) and *Bread Pick Place* (two-stage). These real-world environments included added clutter to simulate more realistic settings. The success rates here were 80% within 30k interactions steps and 95% within 12k interactions steps, respectively, showcasing the framework's adaptability to diverse and unstructured environments.
>
> *R1: "How does trajectory complexity affect the learning cost... How feasible is it for users to sketch accurately for more complex, multi-step tasks?"*
>
> Sketch-to-Skill naturally extends to multi-step tasks through segmented sketching, following the common practice of decomposing complex tasks into simpler steps. As shown in Figures 15, 16, and 18 (Sections E1, E2, and E3), this approach successfully completes complex manipulation sequences including assembly and pick-and-place tasks, demonstrating the same effectiveness as single-step scenarios. For instance, in the MetaWorld Assembly task, our sketch-based approach achieved a success rate of 93%, closely matching the performance of baseline methods which generally average around 95% in similar tasks.
>
> R1: "How sensitive is the model to changes in the scene, such as additional objects or occlusions?"
>
> To address performance in realistic settings, we conducted thorough testing in cluttered environments. We added various daily objects and occlusions in the Toaster Button Press and Bread Pick Place tasks, as shown in Figures 16 and 17. Results demonstrate consistent performance despite environmental complexity.
>
> These comprehensive experiments demonstrate that our framework effectively handles complex, multi-step tasks and performs robustly in realistic, cluttered environments, supporting its practical applicability in real-world robotics applications.
>
> **Discriminator Contribution**
>
> *R1: "The discriminator doesn't appear to contribute significantly... unclear how essential it is or if it's been optimized effectively for generalization"*
>
> We conducted additional experiments to explicitly demonstrate the discriminator's value. In MetaWorld tasks (Figures 5 and 6), we tested 'sketch-to-skill no-BC' scenarios to isolate the discriminator's impact. The results show that 'sketch-to-skill no-BC' consistently performs better than pure RL, demonstrating the discriminator's effectiveness in guiding exploration even without behavior cloning. We further validated this in more challenging scenarios - in Pick and Place Can and Assembly tasks (Figures 15 and 18), where baseline BC scores with sketches were initially low (0.1 and 0.05), incorporating the discriminator loss markedly improved performance. While simpler tasks work well with standard exploration, these results show the discriminator becomes particularly valuable as task complexity increases.

---

> > ### Author Response · Authors · 2024-11-24
> > **Response to Reviewer 1 vobB (2)**
> >
> > **Sketch Perspectives & Implementation:**
> >
> > *R1: "The paper doesn't clarify if there are specific guidelines for choosing sketch perspectives... might need its own camera setup... how does the model manage paths that overlap or are partially obscured?"*
> >
> > Our approach uses a standardized, task-independent camera setup with two orthogonal views to resolve depth ambiguity (line 152). For real robot experiments (Fig. 13), we use two fixed environmental cameras at roughly 90° angles. This configuration works consistently across all tasks without task-specific adjustments.
> >
> > *R1: "Sketching complex trajectories might require a lot more effort than the straightforward examples presented..."*
> >
> > We respectfully disagree. Traditional demonstration methods like kinesthetic teaching and teleoperation require expensive equipment and expert knowledge, while our sketch-based approach only needs basic drawing input, making it significantly more accessible and cost-effective.
> >
> > *R1: "Limited information on how the framework would deal with cases where paths overlap or are partially obscured..."*
> >
> > For overlapping paths, color gradients can be used to represent temporal progression, as illustrated in [3], enabling clear differentiation between trajectory segments. For partially obscured paths, our framework leverages humans’ ability to infer spatial structures despite partial occlusion, as the input sketches are drawn by humans, then we generate the 3D trajectories based on these provided sketches.
> >
> > *R1: "What are the most common sketching errors expected in real-world use? How does the model handle these?"*
> >
> > Common sketching errors include variations in stroke width, unsteady lines, and minor misalignments in start/end points. To handle sketching variations and errors, we employ comprehensive data augmentation during training (detailed in Appendix B), including diverse image augmentations (rotations, scaling, noise) for training the VAE part and controlled elastic deformations to both sketches and trajectories for training the full generator. Our training process explicitly accounts for the variability in hand-drawn sketches while maintaining consistent trajectory generation, helping ensure robustness to common drawing imperfections.
> >
> > [1] Seo, Y., Hafner, D., Liu, H., Liu, F., James, S., Lee, K., & Abbeel, P. (2023, March). Masked world models for visual control. In *Conference on Robot Learning* (pp. 1332-1344). PMLR.
> >
> > [2] Mandlekar, Ajay, et al. "What matters in learning from offline human demonstrations for robot manipulation." *arXiv preprint arXiv:2108.03298* (2021).
> >
> > [3] Gu, Jiayuan, et al. "Rt-trajectory: Robotic task generalization via hindsight trajectory sketches." *arXiv preprint arXiv:2311.01977* (2023).

---

> > > ### Comment · Reviewer_vobB · 2024-11-26
> > >
> > > Thank you for the detailed responses and additional experiments. While the revisions improve the paper and address some of my concerns, several key points remain insufficiently clarified. Regarding overlapping paths and partial occlusions, the proposed use of color gradients and reliance on human inference are reasonable ideas but lack experimental validation. For instance, it remains unclear whether the color gradients are consistently interpretable in more complex trajectories or how the success rate is affected by significant occlusions. Similarly, the standardized camera setup is a good baseline, but further analysis is needed to understand how changes in perspective or task complexity, such as higher-dimensional setups or challenging occlusions, impact the framework's generalization.
> > >
> > > The role of the discriminator is supported by experimental results, particularly in complex tasks, but its generalization to unseen tasks and potential optimization for broader applicability are not fully explored. Additionally, while the data augmentation methods are valuable, the specific impact of common sketching errors, such as line misalignment or overlapping paths, remains unclear, and further data or analysis could provide deeper insights into how these errors influence task performance or failure rates.
> > >
> > > Furthermore, for tasks involving gripper rotation, such as those requiring predictions of full 3D poses (position + rotation), the mechanism for mapping 2D sketches to complete 3D trajectories is not fully explained. While the experiments in tasks like RoboMimic Can Pick and Place show success, there is little discussion on how the model infers rotation from 2D sketches that inherently lack such information. For instance:
> > >
> > > - Does the framework rely on predefined templates or rules to predict rotations?
> > > - Are there additional supervisory signals or dataset requirements to handle rotations effectively?
> > > - How does the model ensure that predicted rotations align with task objectives, especially in cases of asymmetric or complex object manipulations?
> > >
> > > Lastly, while the experiments demonstrate robustness in environments with moderate changes, such as added clutter, they do not address the framework's adaptability to dynamic changes or higher complexity, such as moving objects or more intricate occlusions. These clarifications would provide a stronger foundation for understanding the framework's scalability and robustness in real-world scenarios.

---

> ### Author Response · Authors · 2024-11-27
> **Response to Reviewer 1 vobB (3)**
>
> Thank you for the suggestion. We've conducted some additional experiments to address the concerns about overlapping and complex trajectories. (Additional experiments and evaluation are reported in the appendix in the new version of the paper and marked in red color).
>
> *R1: “ it remains unclear whether the color gradients are consistently interpretable in more complex trajectories…”*
>
> We implemented the notion of color gradients for time-parameterization of the trajectory in the sketches and conducted experiments with overlapping trajectories. These are reported in Appendix E5.   This notion of color gradient in sketches was introduced by RT-Trajectory [1]. An example is shown in Figure 19. Here, the color of the sketch changes from green (start) to red (end). This is particularly helpful when the sketch crosses each other or overlaps. We conduct additional experiments with the generator to evaluate the effect of incorporating gradients in the Sketch-To-3D trajectory generator.
>
> Specifically, we use the MetaWorld Assembly task (Figure 18) where the sketch overlaps in the middle as the end-effector picks up the tool and carries it to the goal position. We trained two generators without and with color gradients. The performance of these two generators are reported in Table 3. We observe that incorporating the gradient in such a case results in lower reconstruction and trajectory losses. With lower trajectory losses we can handle more complex trajectories that overlap with color gradients. Note that incorporating gradients also does not require any change to the downstream architecture, and only requires minimal changes to the generator architecture.
>
> *R1: ”regarding … partial occlusions...reliance on human inference are reasonable ideas…”*
>
> Regarding occlusions: this is part of the reason why we use two camera views. This reduces the risk of both views with significant occlusions, and allows the human to still be able to draw the sketches. In general, we do not need the sketches to be perfect or to include all the finer details. Our key contribution is to show that even with coarse sketches (which may happen with a clutter/occlusions in the environment), we can still better explore in RL with BC bootstrapping and discriminator-based reward shaping. In fact, in some of the real-world experiments, we do observe occlusions in one of the views without affecting performance.
>
> *R1: “The role of the discriminator is supported by experimental results, particularly in complex tasks, but its generalization to unseen tasks…”*
>
> The discriminator in our framework, inspired by standard imitation learning. It evaluates whether a transition resembles those from expert trajectories. Originally, discriminators[2,3]  have been used to evaluate transitions against actual demonstrations from experts.  Our approach innovates by applying the discriminator to transitions from sketch-generated trajectories, leveraging simplified inputs while maintaining critical evaluation.
>
> Training a discriminator typically requires extensive data collection. We address this by generating diverse trajectories from each sketch, reducing the need for numerous demonstrations and improving learning efficiency. Our generator remains task-agnostic, showing generalization across tasks without retraining. The experiments presented in the paper involve tasks unseen during the generator's training.

---

> ### Author Response · Authors · 2024-11-27
> **Response to Reviewer 1 vobB (4)**
>
> *R1: "For tasks involving gripper rotation...Does the framework rely on predefined templates …"*
>
> The framework doesn’t rely on predefined templates or rules to predict rotations. This is the key contribution of our work - we do not need the sketches to be perfect or to include all the finer details. The bootstrapping with BC policy and the discriminator reward shaping act as guidance during exploration in RL improving the learning performance.The rotations are learned from RL interactions with the environment; we obtain reward feedback from the environment to ensure the alignment with task objective.
>
> For example, in the RoboMimic’s Can Pick and Place task, our system successfully learns the necessary grasping angles. This is achieved despite the sketches only providing basic paths for reaching and placing. This demonstrates a key advantage of our approach: we can bootstrap complex task learning from simple 2D sketches, allowing the RL to uncover and master the finer details necessary for task completion. Essentially, the sketches give the robot a rough map of "where to go," while the RL determines "how to get there," refining the approach through active interaction and learning.
>
> *R1: "While the experiments …they do not address the framework's adaptability to dynamic changes or higher complexity, such as moving objects”*
>
> This is a good point. Our current work is focused on static scenarios. Handling dynamic environments is very interesting but beyond the scope of this particular work. Our primary aim is to show that in many cases where we use teleoperated demonstrations, we can use sketches to simplify and expedite the learning process. Addressing dynamic changes involves additional complexities which are interesting but would constitute a separate line of research.
>
> [1] Gu, Jiayuan, et al. "Rt-trajectory: Robotic task generalization via hindsight trajectory sketches." *arXiv preprint arXiv:2311.01977* (2023).
> [2] Zhang, Wenjia, et al. "Discriminator-guided model-based offline imitation learning." *Conference on Robot Learning*. PMLR, 2023.
> [3] Xu, Haoran, et al. "Discriminator-weighted offline imitation learning from suboptimal demonstrations." *International Conference on Machine Learning*. PMLR, 2022.

---

> > ### Comment · Reviewer_vobB · 2024-11-27
> >
> > Thank you for providing the detailed response and additional materials. They addressed many of my questions and provided greater clarity on the paper. While I believe further explainability in the analysis would better highlight the potential and strengthen the overall persuasiveness of the work, I appreciate the effort made to address the concerns. Based on this, I will raise my score.

---

> > > ### Author Response · Authors · 2024-12-02
> > > **Follow-Up on Revised Submission and Additional Experiments : Reviewer 1 vobB (5)**
> > >
> > > Thank you for acknowledging the updates and improvements we made in response to your feedback, and for raising your score based on these changes.
> > >
> > > Given the nearing deadline, we would be grateful if you could consider revising your score to reflect an accept decision, should you feel that all major concerns have been sufficiently addressed. We believe that the enhancements made significantly strengthen the paper and align with the goals of the conference.
> > > We remain open to further discussion and are willing to provide any additional clarifications or information that might assist in your final evaluation. Thank you once again for your constructive feedback and consideration.

---

### Author Response · Authors · 2024-12-03
**General Response**

We extend our sincere thanks to all reviewers for their insightful feedback, which has significantly improved the quality and scope of our work. The new edits are highlighted in blue and red in the updated manuscript. We are particularly encouraged by the recognition of several key aspects of our paper:

1. **Innovative Approach:** Reviewers appreciated the novelty of using human-drawn sketches to streamline data collection, offering a scalable alternative to traditional methods in robot learning. Particularly Reviewer beAX believes our method produced “*Good results on a novel method…removing the need for high-quality teleoperated demonstration data. … method is designed logically*”. whereas reviewer vobB also mentions “*paper introduces an innovative approach…*”, Reviewer ne7n says “*The core idea of generating 3D trajectories…is novel*”
2. **Comprehensive Experiments:** Our thorough evaluation across both simulated and real-world tasks was noted, emphasizing the robustness and applicability of the framework. Reviewer k4pp thinks “*the paper is very thorough with its analysis, and the evaluation does indeed shine a positive light…*”, Reviewer cvP5 appreciated we *“demonstrated on MetaWorld…could help others reproduce the work”*.
3. **Real-World Impact:** The potential of our approach to make robotic learning more accessible by reducing reliance on expensive teleoperation setups and enabling intuitive data collection through sketches was recognized. Reviewer beAX says *“Tests in the real-world show practical applicability of the method*” and Reviewer vobB mentions*“This approach potentially broadens access to robotic training methods.”*

---

### Key Updates During Rebuttal

### Handling Complex Tasks

To demonstrate our framework's scalability to more complex, long-horizon tasks (vobB, k4pp),  we included new experiments on benchmarks like **MetaWorld Assembly** and **RoboMimic Can Pick-and-Place** and **Real-world Bread-Pick-and-Place** (Figure 15, 17 and 18), which require multi-stage planning. These results demonstrate our framework’s ability to generalize and perform effectively on challenging tasks.

### Importance of the Discriminator

Further experiments have highlighted the discriminator's crucial role (vobB, beAX, cvP5), particularly in hard tasks. We demonstrated its significant impact through additional experiments, particularly in hard tasks. For instance, on the Assembly task (Figure 18), success rates improved from **20% to 85%** with the discriminator, underscoring its critical role in guiding RL.

### Robustness of the Sketch-to-3D Generator

In response to queries about the generator’s ability to generalize and handle sketch variations, our ablation studies validated the **VAE-based generator's** effectiveness. It consistently produces smooth, accurate trajectories even from imperfect sketches and adapts well to unseen tasks without retraining (Tables 2 and 3). (ne7n, cvP5)

### Real-World Application

Addressing concerns about real-world applicability (vobB, beAX), we expanded our experiments to include settings with randomized clutter, such as Bread Pick-and-Place and Toaster Button-Press (Figures 17 and 18), demonstrating robust performance in unstructured environments.  Furthermore, as highlighted in our discussions, while sketches do not replace the precise control offered by teleoperation for complex tasks, they provide an accessible and cost-effective means of guiding robotic learning in environments where high precision is not paramount. Our approach effectively leverages human-drawn sketches to initiate learning, broadening the applicability of sketch-based methods in practical robotics scenarios.

### Comparison to Teleoperation

Clarifying the advantages over teleoperation (k4pp, cvP5), we showed that sketches significantly reduce reliance on resource-intensive demonstration collection methods. Our approach achieves **96% of teleoperated performance** and a **170% improvement over pure RL**, offering a scalable and efficient alternative.

---

### Final Remarks

The reviewers have acknowledged the improvements made to our manuscript, with all but one increasing their scores or maintaining an already high score of 8 following our responses. One reviewer maintained their original score, expressing ongoing concerns about the application of sketch-based demonstrations for complex, real-world tasks. Our method builds upon existing research on sketch-based instructions for robots, extending its utility to reinforcement learning environments. While sketch-based methods are not intended to replace high-precision teleoperation for all tasks, they offer a valuable and accessible complement, especially in scenarios where teleoperation is impractical or unavailable. We hope this context and the positive changes most reviewers recognize will be considered in the final decision.

---

### Meta-Review · Area_Chair_evT6 · 2024-12-18

**Metareview:**

The paper presents an approach for combined imitation and reinforcement learning with the scaffolding of sketches. This is a borderline paper with arguments on both the positive and the negative side. On the positive side, the idea of generating 3d trajectories from 2d sketches is very interesting. Further the tricks to improve diversity of trajectories by tuning the noise is interesting as well. On the negative side, there are strong concerns on the applicability of this work for cutting edge robotics problems. After the rebuttal period, the reviewers, with a couple who are experts in robotics, engaged in significant discussion. The following comment summarizes the concerns quite succinctly and I think will benefit the authors as they improve this work for a future submission.

"The main claim that has emerged, it would appear, is that the proposed method can be an easily-accessible complement to teleoperation. This claim, however, is very feeble. If one already has a robot system in place, would (for example) a VR handset device for teleop really be so inaccessible? And if the proposed sketch-based system can only be used for very simple tasks, then what good is it that it's accessible anyhow? If I saw a feasible path towards extending this system for the problems roboticist actually struggle with (e.g. get a robot to fold laundry), then I would be much more positive. But, despite the author's attempts to answer this concern, I remain convinced that the path forward for this work is exceedingly difficult, and it would likely be the case that teleop is still the main way to go in terms of providing robot demonstrations."

**Additional Comments On Reviewer Discussion:**

The most significant concern raised in discussion is the applicability of this work for harder, more challenging robotics tasks.

---

### Decision · Program_Chairs · 2025-01-22

Reject